

# Surface grain-size mapping of braided channels from SfM photogrammetry

Loïs Ribet[1], Frédéric Liébault[1], Laurent Borgniet[2], Michaël Deschâtres[1], Gabriel Melun[3]

[1]Université Grenoble Alpes, INRAE, CNRS, IRD, Grenoble INP, IGE, 38000 Grenoble, France
[2]Université Grenoble Alpes, INRAE, LESSEM, 38000 Grenoble, France
[3]Office Français de la Biodiversité, 94080 Vincennes, France

*Correspondence to*: Loïs Ribet (lois.ribet@inrae.fr)

**Abstract.** Braided channels are known as fluvial systems with a high heterogeneity of physical conditions, resulting from
particularly active interacting processes of coarse sediment sorting and transport. This in turn generates a complex mosaic of
terrestrial and aquatic habitats, supporting an exceptional biodiversity. However, documenting this physical heterogeneity is
challenging, and notably the textural variability of these rivers, which is particularly strong. Distributed and continuous grain-
size maps of braided channels are notably of great interest in this regard. In this study, high-resolution imagery obtained from
UAV equipped for direct georeferencing were used to produce 3D point clouds (Structure from Motion photogrammetry),
from which surface grain-size has been inferred. A set of 12 braided river reaches located in SE of France were used to calibrate
a roughness-based grain-size proxy, and this proxy was used for the production of distributed grain-size maps. The calibration
curve can be used to determine the surface median grain-size with an independent error of 5 mm (14% of relative error).
Resampling procedure shows a good transferability of the calibration, with a residual prediction error ranging from 5 to 17.5%.
Reach-averaged median grain-sizes extracted from roughness-based grain-size maps were in very good agreement with values
collected in the field from intensive grain-size samplings (differences of less than 5%). Some examples of morpho-sedimentary
signatures derived from these maps are provided. They notably show a systematic altimetric gradient of the maximum grain-
size of bars, that is interpreted as an hydrological imprint, that should be better integrated into conceptual models of grain-size
patchiness developed for these rivers.

## 1 Introduction

Unvegetated portions of braided river channels are composed of a mosaic of alluvial bars and threads of heterogeneous surface
grain-size, which reflects not only the flood regime of the catchment  (Storz-Peretz and Laronne 2013; Storz-Peretz et al.
2016), but also the interacting processes of sediment sorting and transport particularly active in these rivers (Bluck, 1979;
Ashmore, 1982, 2013; Gardner et al., 2018). This heterogeneity of surface grain-size conditions, often referred to as the grain-
size patchiness of braided channels (e.g. Guerit et al. 2014), contributes to the diversity of aquatic and terrestrial habitats, and
thus *in fine* to the high degree of biodiversity of these rivers (Ward et al., 1999; Tockner et al., 2003; Dufour et al., 2007; Gray





and Harding, 2007). Surface grain-size distribution (GSD) is also a fundamental parameter of river channels influencing flow resistance and bedload transport, and high-resolution distributed grain-size maps are of uppermost interest for the numerical modelling of braided channel morphodynamics (Williams et al., 2020). It is therefore crucial to develop techniques to continuously characterize in space the surface grain-size patchiness of these complex and unstable channels.

Field sampling methods for surface grain-size measurement, such as the well-known Wolman pebble count based on 100 randomly collected particles (Wolman, 1954), are time-consuming and are mostly used to obtain a representative GSD for a given channel reach or a given sedimentological unit (Bunte and Abt, 2001). Close-range photosieving, which consists in the manual or automatic extraction of a GSD from close-range imagery, is an alternative solution continuously improving since the late 1970s (Adams, 1979; Graham et al., 2005b, a; Buscombe, 2013). These methods are limited by the fact that only

the visible part of grains is measured, so the overlaps between grains, the burial and the 2D projection depending on the view of the photograph drags an underestimation of the grain diameters. Despite these technical bias, photosieving allows saving time in the field with typical measurement errors of 10 % to 20 % (Buscombe, 2013; Chardon et al., 2022) . However, close-range photosieving is not really adapted for producing distributed grain-size maps along kilometer-scale river reaches. Remote sensing approaches for large-scale grain-size mapping emerged from the early 2000s using high-resolution airborne digital

imagery (see Piégay et al., 2020 for a recent review). Image texture and semivariance proved to be successful predictors of the median surface grain-size ($D_{50}$), and continuous grain-size maps were thus produced from 3-cm resolution ortho-images for a 80-km river reach including dry and shallow wetted areas (Carbonneau et al., 2004, 2005), and from 6-cm resolution ortho-images for a 12-km reach including only exposed gravel bars (Verdú et al., 2005).

Another remotely sensed grain-size proxy that emerged in the late 2000s is the channel surface roughness, which can
be captured using dense 3D point clouds. The first application was based on terrestrial laser scanning (TLS) data collected on a 180 m² gravel bar in UK, showing very good correlations between percentiles of roughness height and those of particle diameters computed for the 3 axes (Heritage and Milan, 2009). This dataset demonstrates that surface roughness, computed as twice the standard deviation of elevations, is closer to the particle *c*-axis, and that local calibration curves are site-specific due to imbrication and burial effects. This was confirmed by Hodge et al. (2009) who showed with TLS and grain-size data

collected on two rivers, including the braided River Feshie (Scotland), that grain packing has an effect on the relation between grain-size and the standard deviation of elevation. Subsequent explorations of grain-scale topography with TLS data collected in River Feshie demonstrated that detrending the local relief before computing the surface roughness substantially improves the grain-size calibration curve (Brasington et al., 2012; Rychkov et al., 2012). Another advantage of this approach is that the standard deviation of elevation is computed directly from 3D point clouds, avoiding interpolation effects on elevation statistics.

A similar approach was recently used to produce grain-size maps along four exposed gravel bars of a braided river in New Zealand, using TLS data and a local calibration curve based on 27 pebble counts (Reid et al., 2019). Recent field experiments have shown that roughness-derived surface sedimentology maps can also be produced using Mobile Laser Scanning (Williams et al., 2020) and airborne LiDAR data (Chardon et al., 2020). This latter study conducted in exposed gravel bars of the Rhine





River used not only the roughness, but also the intensity value of the LiDAR signal as a grain-size proxy, with better results with the latter.

The last generation of remotely sensed grain-size mapping approaches is based on the use of Unmanned Aerial Vehicles (UAV) which offer operational flexibility and a more affordable solution than all the other technologies used until now. The first application of drones for surface grain-size mapping was implemented in a 1-km reach of a gravel-bed river in Canada, where texture extracted from a 5-cm resolution ortho-image was used for grain-size extraction on exposed gravel bars using a local calibration curve (Tamminga et al., 2015). Hyperscale images obtained from drones can also be used to produce dense 3D point clouds from SfM photogrammetry (James and Robson, 2012; Westoby et al., 2012), from which surface roughness can be used to map grain-size, following the approach developed with laser scanning datasets. The first reported field experiment for river sedimentology using SfM data showed that a local calibration curve for exposed areas of a braided channel reach can be obtained with SfM point clouds, and this with point density (40-900 points m$^{-2}$) much lower than those obtained with laser scanning (Vázquez-Tarrío et al., 2017). Similar results were obtained earlier with a SfM dataset covering a moraine complex (Westoby et al., 2015). Another experiment in a small stream dominated by cobbles and boulders revealed that 3D topographic data derived from drone images offer a better grain-size prediction than 2D textural patterns in the imagery (Woodget and Austrums 2017). However, when image quality is improved with camera stabilizing gimbals, texture extracted from single drone images offers a better grain-size prediction than roughness (Woodget et al., 2018). Beyond the local demonstration of sedimentological applications from 3D point clouds, a strong variability of empirical relationships emerges from case studies. A series of field and laboratory SfM experiments were designed to explore controlling factors for this variability, and bring insights into the effects of sediment sorting, particle shape, and grain packing in the roughness calibration curve (Pearson et al., 2017). Comparison of roughness metrics derived from 3D point clouds obtained from SfM and TLS surveys of the same gravel bar showed that differences obtained between surveying methods are much smaller than those related to the grid resolution used for surface detrending (Neverman et al., 2019). This illustrates that differences in roughness calibration curves are not only related to local sedimentological properties, but also to data processing protocols.

The most recent innovations in drone applications for grain-size mapping are related to direct georeferencing and to data processing using data-driven machine learning. The robotic photosieving approach based on a low-cost multirotor drone equipped for direct georeferencing allowed the production of undistorted near-ground images scaled with a SfM workflow, and then processed with the photosieving program BASEGRAIN (Detert and Weitbrecht, 2012) to automatically obtain a grain-size distribution (Carbonneau et al., 2018). Field testing of this approach demonstrated that robotic photosieving GSDs are statistically equivalent to those obtained with a traditional SfM workflow, confirming that high-quality grain-size data can be directly obtained from drones without the need of ground control points. However, robotic photosieving implies acquisition of close-range drone images (< 10 m from the ground) and its application domain is spatially limited to short river reaches or small dry exposed areas of river channels. Another recent advance in SfM sedimentology is the data-driven approach for extracting GSD from drone images recently tested on 25 gravel bars along 6 rivers in Switzerland (Lang et al., 2021). A convolutional neural network model calibrated with a training dataset extracted from close-range drone images (10 m flight



height, 0.25 cm resolution) was successfully used to extract the full GSD and characteristic mean diameters of gravel bars. Although the good performance of this approach is conserved up to image resolutions around 1-2 cm, the quality of grain-size

mapping products with such approach remains to be tested with high-elevation drone surveys more suitable for covering kilometer-scale river reaches. Data-driven approach for grain-size mapping in river channels can also be applied to airborne LiDAR data, as recently demonstrated in a 37-km reach of a gravel-bed river (Díaz Gómez et al., 2022).

This paper explores the grain-size patchiness of 12 braided gravel-bed rivers of SE France using SfM 3D point clouds derived from high-resolution imagery obtained from a drone equipped for RTK direct georeferencing. This technological

innovation is expected to improve the quality of SfM 3D point clouds while saving time in the field by reducing the need for ground control points (Hugenholtz et al., 2016; Grayson et al., 2018; Chudley et al., 2019), but it has never been tested for sedimentological applications. Specific objectives are (1) to evaluate the overall performance of automatic photosieving from close-range imagery for field calibration; (2) to evaluate the overall performance of the surface grain-size prediction derived from SfM grain-scale topography obtained with RTK direct georeferencing; (3) to evaluate the transferability of the grain-size

prediction for braided fluvial environments; and (4) to explore applications of the method for mapping and characterizing the grain-size patchiness of braided rivers.

## 2 Methodology

### 2.1 Study sites

The study sites are composed of 12 well-preserved braided river reaches located in the Southern French Alps and Prealps

(Figure 1A; Table 1). Spatial extents (kml file) of the 12 study reaches are available as supplementary material. Drainage areas vary from 9 to 876 km² and active channel widths from ~20 to ~200 m. The selected reaches exhibit typical morphological and sedimentological features of alpine braided channels, with multiple low-flow channels separating unvegetated gravel bars of varying size, shape, and surface texture (Figure 2). Active channels are composed predominantly of gravels and cobbles, with local patches of fine sediment deposition (sands and silts) (Figure 2C). Local concentrations of small boulders are also

present on proximal sites (drainage areas less than ~100 km²; Figure 2D). Most of the study sites are located in the subalpine sedimentary domain, with limestone rocks constituting the dominant lithology of the bed-material. Only two sites are draining the crystalline domain of the inner Alps, in the Ecrins Massif (Séveraisse, Torrent de Saint-Pierre). The bed-material is dominated here by granite and gneiss rocks. The Drac is an hybrid site characterized by more contrasted lithological conditions, with c.a. 80 % of the catchment in sedimentary rocks (mostly sandstones), and c.a. 20 % in the crystalline domain (mostly

gneiss, granite and migmatite).

The 12 study sites have been chosen to explore two gradients influencing the grain-size patchiness of braided channels. The first is the catchment-size gradient controlling the downstream fining of river channels through size-selective transport and abrasion processes. The second gradient is related to the sediment regime, from transport-limited to supply-limited regimes, which can be roughly assessed with the normalized active channel width, computed as $W^* = W / A_d^{0.44}$, with



$W$ the active channel width (in m), and $A_d$ the drainage area (in km²). A regional analysis of braiding conditions in SE France has shown that this metrics can be considered as a good proxy of the sediment regime of braided rivers, with increased normalized width towards transport-limited regimes (Piégay et al., 2009; Liébault et al., 2013). The study sites can be ranked according to their residual from the regional scaling law, with positive residuals for transport-limited regimes, and negative residuals for supply-limited regimes (Figure 1B). A coarser GSD is expected for threads (low-flow channels) of supply-limited

regimes, as compared to transport-limited ones, under vertical size sorting effects.

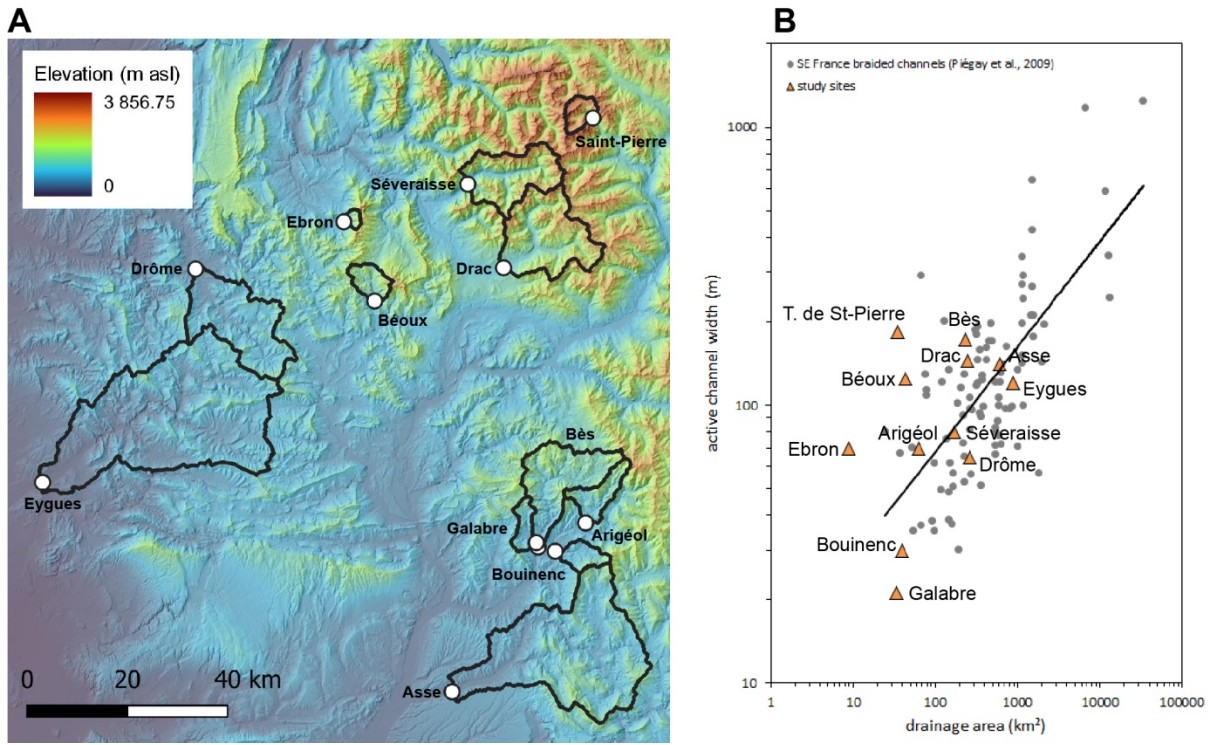

**Figure 1: (A) General map of the study sites, with catchment divides in black; (B) ranking of the study sites with respect to the regional relationship of active channel width vs. drainage area for SE France braided channels (from Piégay et al. 2009).**


| Study sites | Drainage area (km²) | Catchment elevation /mean-min-max (m) | Petrographic dominance in active channel | Active channel width (m) | $W^*$ | Channel slope (m/m) | Length of study reach (m) |
|---|---|---|---|---|---|---|---|
| Arigéol* | 63 | 1386-841-2796 | limestones | 70 | 11.3 | 0.020 | 980 |
| Asse* | 617 | 1016-396-2281 | limestones | 140 | 8.3 | 0.010 | 1900 |
| Béoux* | 43 | 1604-1012-2676 | limestones | 124 | 23.7 | 0.037 | 1735 |
| Bès* | 230 | 1347-656-2736 | limestones, cargneule, sandstones | 172 | 15.7 | 0.010 | 2410 |
| Bouinenc* | 40 | 1134-682-2265 | limestones, marls | 30 | 5.9 | 0.010 | 425 |
| Drac** | 248 | 2030-1070-3415 | gneiss, sandstones, granites | 144 | 12.7 | 0.010 | 2020 |
| Drôme* | 260 | 961-475-1754 | limestones | 65 | 5.6 | 0.010 | 910 |



| | | | | | | | |
|---|---|---|---|---|---|---|---|
| Ebron* | 9 | 1618-1013-2592 | limestones | 70 | 26.6 | 0.080 | 900 |
| Eygues* | 876 | 742-171-1750 | limestones | 120 | 6.1 | 0.005 | 1680 |
| Galabre* | 34 | 1139-672-1848 | limestones, gypsum | 21 | 4.5 | 0.020 | 300 |
| Séveraisse** | 173 | 2095-938-3634 | gneiss | 80 | 8.3 | 0.020 | 1120 |
| Saint-Pierre*** | 35 | 2888-1849-4068 | gneiss, granites | 183 | 38.3 | 0.040 | 1800 |

**Table 1: Main physical features of the 12 study sites; *W*\*: normalized active channel width; \*Mediterranean flow regimes; \*\* snowmelt flow regimes; \*\*\* nivo-glacial flow regime**

A survey reach has been selected for each site, with a length approximately equal to 15 times the mean active channel width. The location of these reaches has been chosen to avoid as much as possible local morphological effects related to common human pressures, such as embankments. The main reason is related to an objective of comparing morphological signatures of undisturbed braided channels, which is not addressed in the present paper. Field surveys took place during low-flow conditions: in autumn for snowmelt and glacial-melt rivers, and in late spring and summer for Mediterranean rivers.

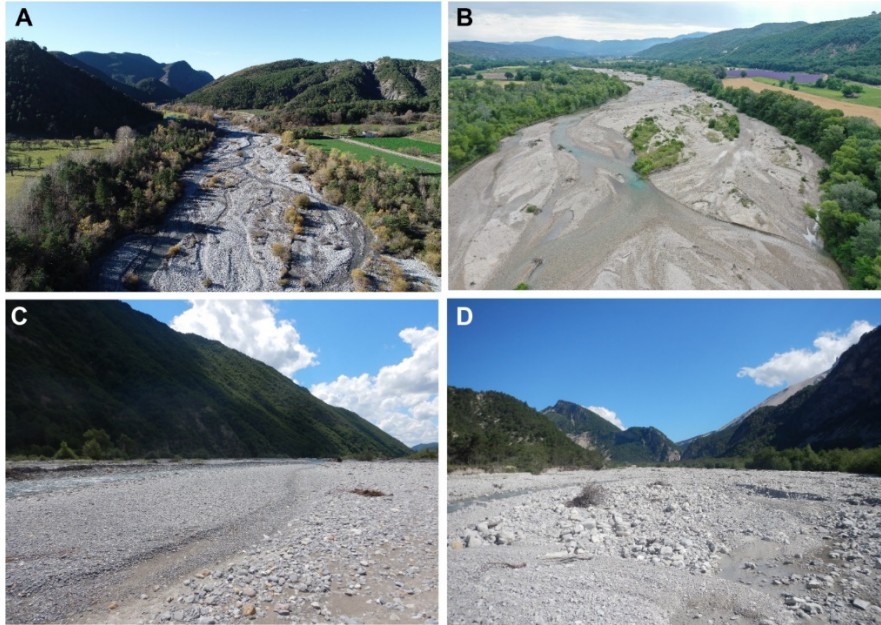

**Figure 2: Pictures illustrating typical morphological and textural patterns of the study sites: aerial (drone) views of the Arigéol (A) and Asse (B) braided channels; surface grain-size patchiness of the Bès (C) and Béoux (D) braided channels; views looking upstream**

## 2.2 Grain-size calibration dataset from close-range imagery

Calibration of the grain-size proxy derived from 3D point clouds was obtained from close-range photosieving using a set of images collected in the field with a drone flying at a very low relative elevation from the ground. Field sampling and data processing procedures of this calibration dataset are presented here.



### 2.2.1 Field sampling of surface grain-size

For each of the 12 study reaches, the dominant grain-size patches of exposed (dry) areas of active channels were identified and
sampled in the field by square plots of two different sizes (Figure 3). A grain-size patch is defined here as a channel portion with a homogeneous surface GSD covering an area > 1 m². A one square-meter plot (100 cm x 100 cm) was used for patches dominated by gravels and pebbles (Figure 3A), while a four square-meter plots (200 cm x 200 cm) was used for patches dominated by cobbles (Figure 3B). About ten plots were sampled per site, giving a total number of 129 plots. Contours of each sampling plot were marked with solvent-free spray paint in order to be detected and extracted on SfM point clouds. A wooden
frame was used for marking the plots, and a rule was placed along the side of the frame for scaling images. The surface of each plot was carefully cleaned by removing vegetation (herbaceous plants and seedlings) and small woody debris before taking pictures. Plots were systematically positioned on flat and homogeneous surfaces, to avoid artifacts related to local relief and poorly-sorted sediment mixtures.

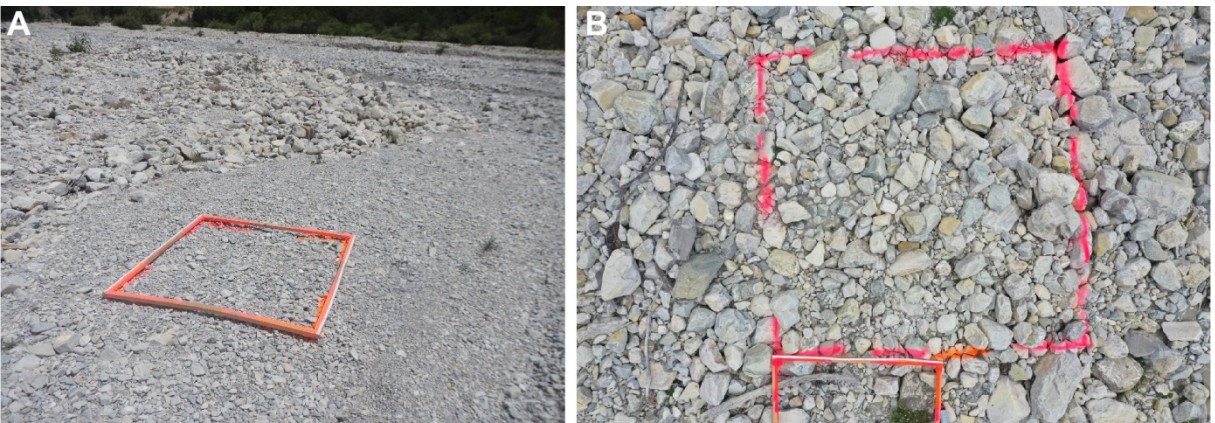


**Figure 3: Examples of sampling plots used for the field calibration of grain-size proxy derived from 3D point clouds: (A) 1-m² plot used for sampling a patch dominated by gravels, Béoux site; (B) 4-m² plot used for sampling a patch dominated by cobbles and small boulders, Béoux site; the wooden frame used for marking the plots, and the rule used for scaling images are visible on each picture.**

Plots were photographed individually by a DJI Mavic 2 drone at a height of 5 to 8 m from the ground. Obtained images have a spatial resolution comprised between 0.22 and 2 mm. Grain-sizes smaller than the spatial resolution are not visible from imagery, and GSDs obtained from close-range photosieving must be considered as truncated at their lower tail, at ~2 mm, which corresponds to the upper size limit of very coarse sands.

### 2.2.2 Photosieving with Digital Grain-size

Close-range images were processed with the Digital Grain-size (DGS) code developed by Buscombe (2013), which automatically provides an estimate of the statistical GSD from an image. The apparent GSD in the image is estimated by



deriving the global spectral density power function using Morlet wavelets. The use of Morlet wavelets allows the simultaneous quantification of spatial and spectral information by decomposing the image into variance versus frequency. This method has the advantage of being fast and free of a calibration phase, which theoretically gives it a universal scope. DGS processing of
images provides the grid-by-number frequency of grains as a function of their diameter as well as the GSD percentiles. A normalized root mean square error ($nRMSE = RMSE/D_x$, with $D_x$ the percentile of rank $x$) for the $D_{50}$ typically less than 20 % can be achieved when the sample size is at least 250 grains per image (Buscombe, 2013).

Following recommendations of Chardon et al. (2020) who tested the performance of DGS photosieving on exposed gravel bars of the Rhine River, images were corrected by a median filter with a radius of 5 % of the largest particle sizes. This
correction attenuates errors related to intra-grain petrographic variations, and allows reducing the $D_{50}$ $nRMSE$ from 72 % to 29 %. This improvement was confirmed by our dataset, and only data from filtered images are presented here. The study by Chardon et al. (2020) also showed that solar lighting conditions and sampling area have an effect on DGS performance. It has been therefore necessary to produce a calibration dataset for controlling and calibrating DGS-derived GSDs.

**2.2.3 Calibration of DGS grain-size percentiles with ImageJ**

Forty-four plots were sampled for testing the quality of GSDs obtained from DGS (10 from the Galabre, 11 from the Arigéol, 3 from the Torrent de Saint-Pierre, 9 from the Ebron, and 11 from the Eygues). The open-source image processing software ImageJ (https://imagej.nih.gov/ij) was used for the manual extraction of apparent diameters visible on the selected images. This software allows the positioning of a grid on the images, the manual segmentation of the grains and the extraction of *a*- and *b*-axes. At least one hundred grains were measured on each image, from which GSD percentiles were obtained. These
manual grain-size measurements were used to verify and calibrate the automatic results produced by DGS. GSDs from ImageJ were considered as *pseudo* ground-truth data, since only apparent diameters can be extracted from images, which are different from true diameters that can be directly measured in the field using a classic pebble count. This is known as the fabric error, which is related to individual grain inclination and partial hiding (Graham et al., 2005a). However, several studies demonstrated that this systematic error is low for grid-by-number sampling, typically comprised between 0.05 and 0.45 $\psi$ ($\psi = \log_2 D$, with
$D$ the grain diameter in mm) (Graham et al., 2005a; Dugdale et al., 2010). Since sediment patches from our study sites are composed of loose grains without strong imbrication and packing, we assumed that the systematic bias from true percentiles is low (likely less than 0.45 $\psi$).

Calibration of DGS grain-size percentiles were obtained by regression analysis with manually extracted percentiles from ImageJ. The root mean square error (*RMSE*), the mean absolute error (*MAE*) and the irreducible random error (*e*) of DGS
percentiles (for both raw and calibrated values) were calculated respectively as:

$$RMSE = \sqrt{\frac{\sum_{i=1}^{n}\left(D_{xpi} - D_{xoi}\right)^2}{n}} \tag{1}$$



$$MAE = \frac{1}{n}\sum_{i=1}^{n}\left|D_{xpi} - D_{xoi}\right| \tag{2}$$


$$e = \sqrt{RMSE^2 - MAE^2} \tag{3}$$

with $D_{xpi}$ the DGS percentile of rank $x$, $D_{xoi}$ the manually extracted ImageJ percentile of rank $x$, and $n$ the number of plots used for calibration. The normalized *RMSE* and *MAE* were also computed to aid comparisons with previous studies (e.g. Chardon

et al., 2020).

**2.3 Grain-size proxy from 3D SfM point clouds**

**2.3.1 UAV surveys**

UAV images were taken with a DJI Phantom4 RTK. This drone benefits from direct georeferencing. It is connected to a D-RTK mobile station, allowing drone positioning with an announced accuracy of 1 cm in $x$ and $y$, and 1.5 cm in $z$ (1-mm error

increase by km of distance to the base). The onboard camera resolution is 20 Mpxl, with a focal length of 8 mm. The flight height was fixed at 70 m allowing a spatial resolution of 2-3 cm on images. Images were taken during low-flow period at nadir with a side and forward overlaps of 60 % and 80 %, respectively. Parameters of the drone flights were programmed with the DJI Ground Station RTK application.

Ground control points (GCP) were marked homogeneously in each site with solvent-free spray paint. They were

positioned along cross-sections spaced at a regular interval equal to the mean active channel width. Their coordinates were measured either with a Leica Zeno 20 dGPS, or with a GPS Leica GS20 and a rover Leica GS10. The Leica Zeno 20 provides corrected GNSS RTK positions, and benefits from the coverage of the GPS, Glonass and Galileo networks. The measurement accuracy announced by the manufacturer is less than 5 cm + 1 ppm in horizontal and less than 2 cm + 1 ppm in vertical (Leica Geosystems, 2015). The GPS Leica GS10 provides a position with an *RMSE* of 8 mm +1 ppm in $x$ and $y$ and 15 mm +1 ppm

in $z$ in cinematic mode and 3 mm +0.5 ppm in $x$ and $y$ and 5 mm +0.5 ppm in $z$ in static mode.

**2.3.2 SfM photogrammetry and roughness extraction**

A SfM photogrammetric processing of UAV images was performed using the Agisoft Metashape software (version 1.7.3) to produce a dense point cloud, a Digital Surface Model (DSM) and an ortho-image. The SfM workflow was built following recommendations available in the literature (Eltner et al., 2016; James et al., 2020; Over et al. 2021) to get the best outputs

regarding the performance of the computer. The sparse points clouds were obtained by the alignment of the photos using a high accuracy. Poor quality points were removed, Ground Control Points were imported, and cameras were optimized several times. The dense points clouds construction was done by setting high quality and mild depth filtering. The workflow is presented in detail in the supplementary material, with different metric of errors.



Metric errors (mean error, mean absolute error and standard deviation of error) of the camera location for the *z* coordinate are automatically calculated by Agisoft Metashape during the reconstruction of the 3D model (*Alignment* tool) and the densification of the cloud. Results show a good accuracy of the dense point clouds with a standard deviation of error for the *z* coordinate of the camera ranging from 0.98 cm (Galabre) to 4.6 cm (Drac). Errors computed with control points (dGPS markers used to georeference the 3D model) and check points (dGPS markers not used to optimize the camera alignment) are also provided in supplementary material (Tab. S1). *RMSE* of elevations computed on check points are generally less than 10

cm, except for 4 sites (Asse, Drac, Drôme, and Ebron). The mean point density of all the dense point clouds is 973 points/m² and the resolution of ortho-mosaic is ranging from 1.27 to 1.54 cm.

| Study sites | Dense point cloud density (pts/m²) | MNS resolution (cm/pxl) | Ortho-mosaics resolution (cm/pxl) |
|---|---|---|---|
| Arigéol | 976 | 2.97 | 1.48 |
| Asse | 897 | 3.04 | 1.52 |
| Béoux | 1189 | 2.90 | 1.45 |
| Bès | 1324 | 2.70 | 1.35 |
| Bouinenc | 1214 | 2.70 | 1.35 |
| Drac | 1464 | 2.54 | 1.27 |
| Drôme | 905 | 3.07 | 1.54 |
| Ebron | 1527 | 2.71 | 1.35 |
| Eygues | 983 | 2.98 | 1.49 |
| Galabre | 613 | 2.79 | 1.40 |
| Séveraisse | 1054 | 2.98 | 1.49 |
| St-Pierre upstream | 980 | 3.04 | 1.52 |
| St-Pierre downstream | 973 | 2.90 | 1.45 |

**Table 2: Resolution and point density of photogrammetric outputs from Agisoft Metashape**

SfM point clouds were imported in CloudCompare (version 2.12.0) (https://www.danielgm.net/cc/) to manually segment the sampling plots at each site and compute their roughness height. This metric is defined as the distance between a given point and the best fitting plane calculated on its nearest neighbors included in a sphere of predetermined radius (CloudCompare, 2021). It is now regularly used as a grain-size proxy (Vázquez-Tarrío et al., 2017; Woodget and Austrums, 2017; Chardon et al., 2020). The radius for calculating roughness height was set at 0.5 m. This value corresponds to the optimal

radius obtained by Vázquez-Tarrío et al. (2017) in a braided channel, and it represents approximately twice the largest grain observed in the dataset. Percentiles of the roughness height distributions were used as predictors of corresponding grain-size percentiles ($D_{16}$, $D_{50}$, and $D_{84}$). The mean roughness height was also used as a predictor for the $D_{50}$. All these predictors were compared to grain-size percentiles derived from close-range photosieving, through regression analysis.

### 2.3.3 Jackknife cross-validation of the grain-size proxy

Validation of grain-size calibration curves was done using a LOOCV (Leave-One-Out Cross-Validation) jackknife procedure (Quenouille, 1956; Woodget and Austrums, 2017). This iterative procedure was used to compute linear regression fits using



128 of the 129 sampling plots and predict grain-size of the excluded plot. This procedure was reiterated 129 times to independently assess accuracy and precision of each calibration curve. Another jackknife procedure was used to assess the transferability of the calibration curves to sites that have not been used for calibration. It was performed by excluding all the plots from one given site. The grain-size of that site is then predicted with the calibration curve obtained from the other sites to evaluate the transferability. This procedure has been reiterated 12 times so that the error of transferability has been computed for each site.

### 2.3.4 Active channel grain-size maps

The $D_{50}$ calibration curve was used to produce a distributed grain-size map of active channels for each study site. After creating a mask of the active channel including unvegetated gravel bars and low-flow channels, non-alluvial elements (large woody debris, artificial objects) were manually removed. Small vegetated patches included in the active channel were filtered with the automatic classification tool available in Agisoft Metashape (*Classify ground points tool*). An in-house R script based on the Excessive Greenness index (ExG) (Woebbecke et al., 1995 ; Núñez-Andrés et al., 2021) was written to remove unfiltered vegetated patches from Metashape. Steep surfaces of the active channel (e.g. bar talus) were also excluded, since they are characterized by high roughness values not related to grain-size but to the local relief. A slope threshold of 60 % was used to remove these steep surfaces, except for the Asse where it has been necessary to use a threshold of 30 % to exclude bar talus. No slope threshold was used for the Ebron, because its active channel presents a coarse surficial grain-size without any clear effect of local slope on roughness heights.

Filtered dense point clouds were imported in CloudCompare to compute the roughness height using a 0.5 m radius. A 1-m resolution raster of roughness was produced by averaging roughness values for each pixel. Finally, the calibration curve was applied on every pixel to get the corresponding $D_{50}$. Each pixel was then included into a grain-size class using the Wentworth scale (Wentworth, 1922), and considering half-phi and phi size limits for gravels and cobbles, respectively. Relative proportions of grain-size classes were then compared between sites, with respect to drainage areas.

A specific field sampling was undertaken to determine if grain-size maps can provide accurate reach-scale estimates of the mean surface $D_{50}$ of the active channel, and of different geomorphic units of the active channel (unvegetated bars and low-flow channels). This was done for the Arigéol and the Drac, from Wolman pebble counts along 10 cross-sections spanning the whole active channel, established at 100 m (for Arigéol) and 250 m (for the Drac) spacing, with a sampling interval of 1 m and 2 m for the Arigéol and the Drac, respectively (cross-sections maps are provided in the supplementary material, Fig. S1 and S3). Sampling points located in bars and low-flow channels were informed during the field sampling, in order to compute composite GSDs of these two geomorphic units. The assemblage of all sampling points were used to compute the active channel GSD. A similar field sampling was initialy planned for a third site (Bouinenc), but for logistical reasons, Wolman pebble counts were restricted to a single point bar where particles were collected at 1-m intervals along two longitudinal





sampling lines (Fig. S2). Confidence intervals (95%) of the $D_{50}$ obtained from field samplings were computed using the GSDtools package developed by Eaton et al. (2019).

Field samplings were done in early 2024, and were compared with grain-size maps extracted from UAV images taken in November 2020 (Arigéol and Bouinenc) and March 2024 (Drac). No major morphological changes occurred between 2020 and 2024 along the Arigéol study reach, and it is assumed that the reach-scale averaged surface GSDs remained unchanged during this period. The investigated point bar of the Bouinenc remained also quite stable during this period. This is not the case of the Drac study reach, which has been modified by an active hydrological period that occurred in October 2023 (peak

discharge with a 10 year return period). Therefore, field-based GSDs were compared to a grain-size map extracted from UAV images taken the same day as the field survey (March 2024). A comparison was also made with a grain-size map extracted from images of September 2021.

The extraction of reach-averaged $D_{50}$ of unvegetated bars and low-flow channels from grain-size maps was done from a manual digitizing of bars using SfM DEMs and ortho-images (see supplementary material for an example, Fig. S4).

Two procedures were used to select bar pixels for grain-size extraction. The first one considers all the pixels totally included in bar polygons. The second one is based on a selection of bar pixels considered as valid. A 4 m² sampling grid (i.e. 4 grain-size pixels) was superimposed on the ortho-image to carry out a selection based on a qualitative inspection of the bar surface. Only grid cells with at least 90 % of the surface composed of clean coarse (gravel, cobble) alluvial sediment were considered as valid. This allows to exclude bar pixels where surface roughness is partly controlled by woody debris or fine sediment

deposits (silts), and thus to include only those pixels that are similar to the sampling plots used to calibrate the roughness proxy. The low-flow channel $D_{50}$ was extracted from pixels that are not included in digitized bars. Finally, the active-channel $D_{50}$ was obtained by averaging bar and low-flow channel pixels.

## 3. Results

### 3.1 Photosieving with DGS

Grain-size percentiles obtained from close-range automatic photosieving with DGS were compared with those obtained from manual extraction of apparent particle diameters with ImageJ (Figure 4). DGS performance was thus tested for a wide range of grain-sizes (8 mm < $D_{50}$ < 154 mm), representative of the sedimentological variability of study reaches. A systematic under-estimation of low percentiles ($D_{10}$ and $D_{16}$) was observed, while high percentiles ($D_{84}$ and $D_{90}$) were consistently over-estimated. Medians of the computed distributions are closer to the equality line, except for two points with a manually extracted

$D_{50}$ above 120 mm for which a strong (~35 %) under-estimation was observed.



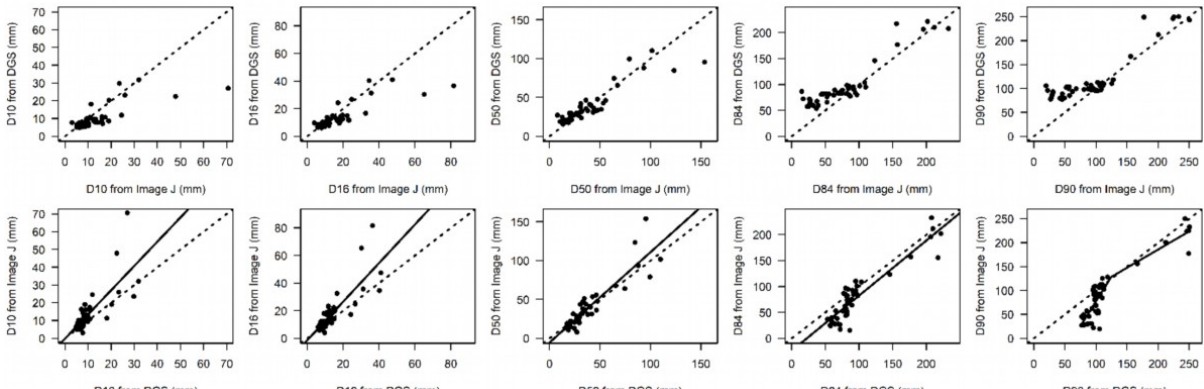

**Figure 4: Comparison of grain-size percentiles computed with DGS and extracted with ImageJ; plots on the top show predicted vs. observed percentiles; plots on the bottom show the calibration curves of the DGS percentiles (full line). Dotted lines correspond to equality lines (x=y)**


The DGS performance for $D_{50}$ prediction is not as good as expected, with a *nRMSE* of 33 %, a *nMAE* of 21 %, and an irreducible error of 10.37 mm (or 3.37 $\psi$) (Table 3). The best performance was obtained for the $D_{50}$ and the $D_{84}$. If the two outliers with a manually extracted $D_{50}$ above 120 mm are excluded, *nRMSE* and *nMAE* for the $D_{50}$ fall at 24 % and 18 %,

respectively. Those outliers correspond to two large sampling plots established on well-sorted cobble bars of the Ebron site. Although the number of grains on those two images is low, it stays close to the recommended limit of 250 grains per image below which DGS performance strongly drops.

Linear regressions offer the best fits for $D_{10}$, $D_{16}$, $D_{50}$, and $D_{84}$ calibration curves (Figure 4). However, normalized errors of calibration stay above 45 % for the lower percentiles, and a value of 32 % is obtained for the $D_{50}$. The best calibration

curve for the $D_{90}$ was obtained using a piecewise regression function allowing the automatic detection of breakpoints in the regression line (Muggeo, 2008). This regression model is justified by an evident break of slope visible on the scatterplot of the highest percentile, revealing that a linear model is not really appropriate for this dataset. A piecewise function was also tested for the $D_{84}$, but without any success because the change of slope above the suspected breakpoint is too low (points with ImageJ derived $D_{84}$ above 120 mm). The best calibration curves are finally those obtained for high percentiles, with *nRMSE* around

345 25 %.

| Error metrics | $D_{10}$ | $D_{16}$ | $D_{50}$ | $D_{84}$ | $D_{90}$ |
|---|---|---|---|---|---|
| *RMSE (mm)* | 8.86 | 10.03 | 13.40 | 27.23 | 35.34 |
| *nRMSE (%)* | 0.63 | 0.56 | 0.33 | 0.34 | 0.37 |
| *MAE (mm)* | 4.76 | 5.52 | 8.48 | 22.10 | 27.23 |
| *nMAE (%)* | 0.34 | 0.31 | 0.21 | 0.27 | 0.29 |
| *e (mm)* | 7.48 | 8.37 | 10.37 | 15.91 | 22.53 |
| *RMSE* of calibration (mm) | 7.71 | 8.43 | 12.81 | 20.43 | 23.59 |
| *nRMSE* of calibration (%) | 0.55 | 0.47 | 0.32 | 0.25 | 0.25 |





**Table 3 : Error metrics of DGS percentiles obtained on close-range grain images after application of a median filter; the root mean square errors of calibration curves obtained for the different percentiles are also indicated (see section 2.2.3. for definitions of the metrics)**


## 3.2 Grain-size from SfM 3D point clouds

### 3.2.1 Calibration curves based on roughness height

Calibration curves obtained for grain-size percentiles ($D_{16}$, $D_{50}$, $D_{84}$) using corresponding roughness height percentiles ($Rh_{16}$, $Rh_{50}$, and $Rh_{84}$) and the mean roughness height ($Rh$) are presented in Figure 5 and Figure 6, respectively. Regression equations

and parameters are displayed in Table 4 along with metric errors computed by the jackknife method. Linear regressions systematically offer the best fits for every tested proxy, and the best calibration curve was obtained for the $D_{50}$ as a function of the mean roughness height ($R^2 = 0.83$). This calibration curve shows an independent error of prediction of 4.97 mm, which corresponds to 14.4 % of the mean $D_{50}$ computed for the 129 calibration plots (34.47 mm). Generally speaking, and whatever the considered grain-size percentile, better results are obtained with the mean roughness height, compared to roughness

percentiles. Better results are also obtained for $D_{16}$, compared to $D_{84}$. The high data scatter observed for the $D_{84}$ diagram shows that the roughness height is not a good proxy of the coarse surficial grain-size.

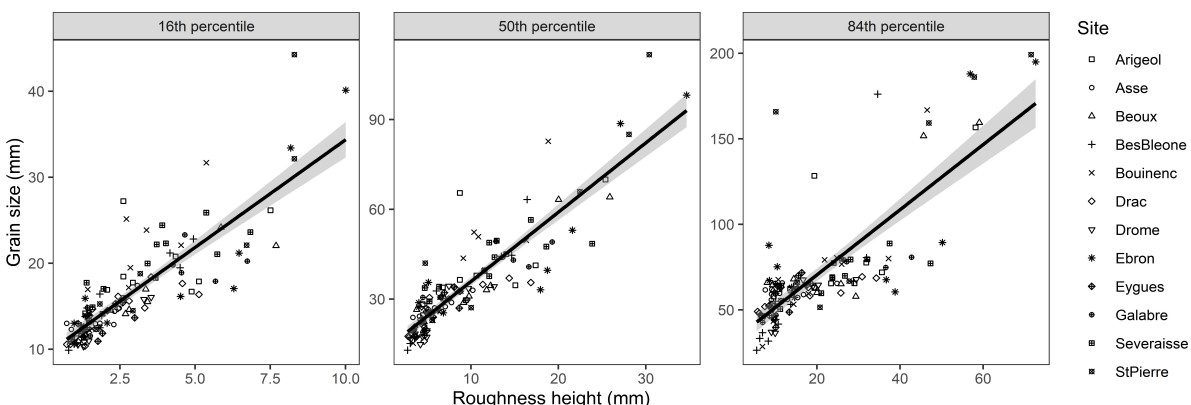

**Figure 5: Calibration curves of grain-size percentiles based on corresponding roughness height percentiles. Shaded areas correspond**
**to the 95% confidence interval of the regressions.**



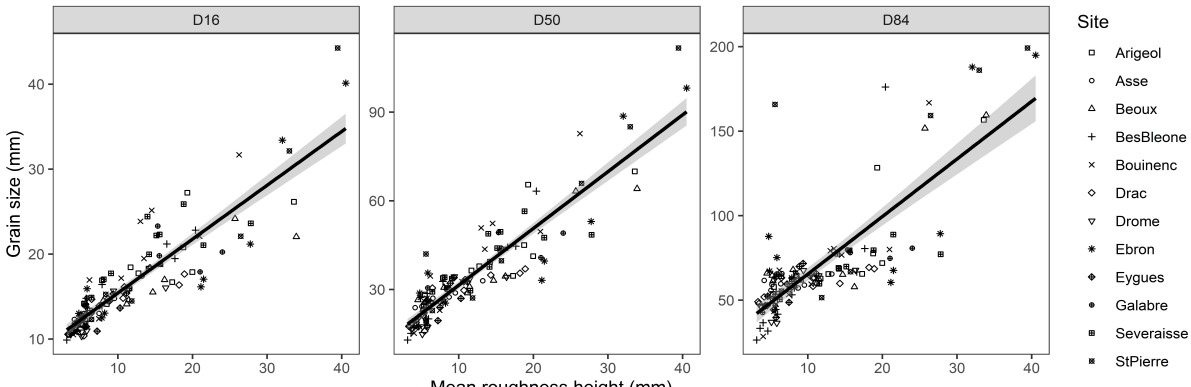

**Figure 6: Calibration curves of grain-size percentiles based on the mean roughness height. Shaded areas correspond to the 95% confidence interval of the regressions.**


| Linear regressions | $R^2$ | $p$-value | Residual errors (mm) | |
| --- | --- | --- | --- | --- |
| | | | Mean | Standard deviation |
| $D_{16} = 2.5\ Rh_{16} + 9.3$ | 0.72 | < 0.0001 | 2.15 | 2.34 |
| $D_{50} = 2.3\ Rh_{50} + 13$ | 0.79 | < 0.0001 | 5.53 | 5.89 |
| $D_{84} = 1.9\ Rh_{84} + 32$ | 0.62 | < 0.0001 | 14.32 | 16.31 |
| $D_{16} = 0.63\ Rh + 9.1$ | 0.79 | < 0.0001 | 1.90 | 2.03 |
| $D_{50} = 1.9\ Rh + 12$ | 0.83 | < 0.0001 | 5.35 | 4.97 |
| $D_{84} = 3.4\ Rh + 31$ | 0.64 | < 0.0001 | 14.12 | 15.91 |

**Table 4: Linear regressions of grain-size percentiles calibration curves with their Jackknife residual errors**

### 3.2.2 Transferability of the $D_{50}$ calibration curve

To test the transferability of the $D_{50}$ calibration curve to other braided rivers, a jackknife resampling was undertaken to evaluate
expected deviations from the curve once applied to a site not included in the calibration. Results are shown in Figure 7. For each diagram, the calibration curve was established after excluding all the plots from one given site. Points in each diagram correspond to the plots of the excluded site. The mean error, the normalized mean error and the normalized standard deviation of error specific to each site were calculated in order to quantify accuracy and precision (Table 5). The normalization was made using the mean $D_{50}$ of each site. The residual prediction error varies from 5 to 17.5 % (Asse and St-Pierre, respectively).
This result is in good agreement with the independent prediction error of the $D_{50}$ calibration curve. Asse is the site for which points are closer to the calibration curve, with the lowest normalized mean error. However, calibration curves tend to overestimate the median grain-size of Béoux, Drac, and Drôme, and to underestimate those of Bouinenc and St-Pierre.





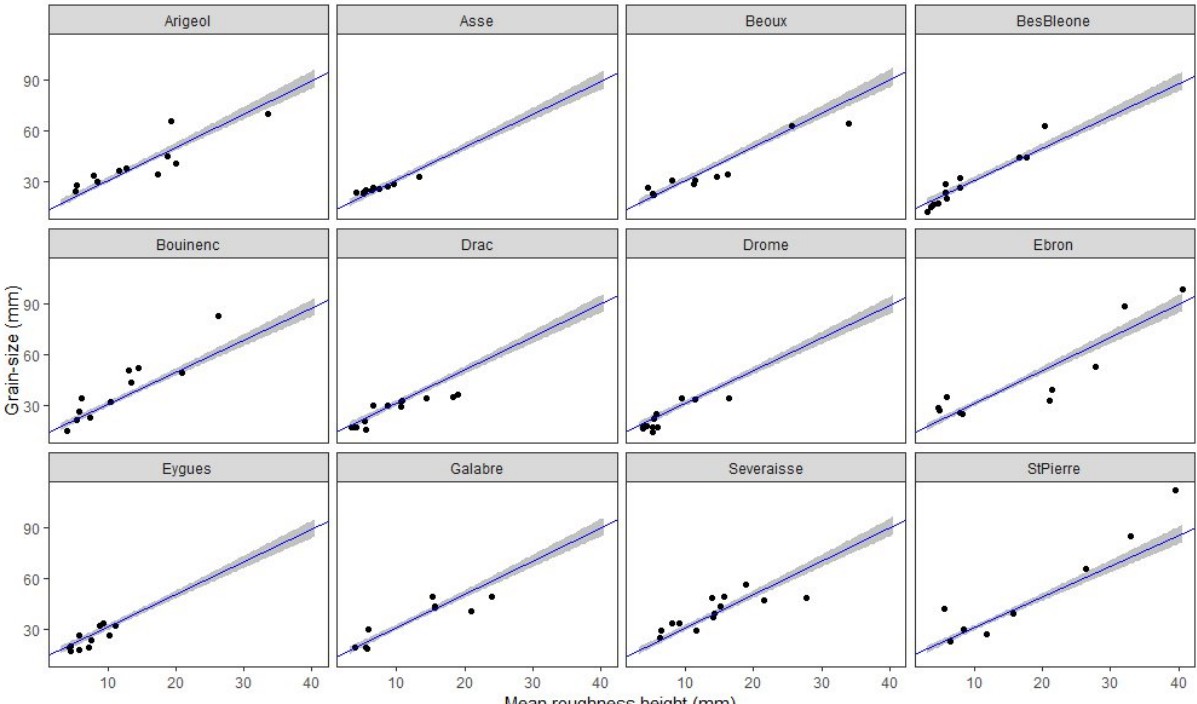

**Figure 7: Assessment of the transferability of the $D_{50}$ calibration curve. For each plot, linear regressions were obtained after exclusion of calibration plots from the mentioned study site. Points represent calibration plots of the excluded site. Shading area represents the 95% confidence interval of the regression.**

| Study Site | Mean error (mm) | Normalized Mean Absolute Error | Normalized Standard Deviation of the Error |
|---|---|---|---|
| Arigéol | 0.335 | 0.150 | 0.115 |
| Asse | 0.513 | 0.084 | 0.048 |
| Béoux | -3.042 | 0.142 | 0.128 |
| Bès | 0.066 | 0.133 | 0.110 |
| Bouinenc | 5.295 | 0.185 | 0.165 |
| Drac | -3.172 | 0.152 | 0.151 |
| Drôme | -2.881 | 0.170 | 0.129 |
| Ebron | -0.344 | 0.215 | 0.129 |
| Eygues | -1.652 | 0.134 | 0.080 |
| Galabre | -1.419 | 0.145 | 0.119 |
| Séveraisse | 1.108 | 0.143 | 0.113 |
| St-Pierre | 6.811 | 0.181 | 0.175 |

**Table 5: Metrics of error of the $D_{50}$ obtained from the jackknife resampling procedure based on site exclusion (transferability analysis)**



### 3.3 Surface grain-size mapping

### 3.3.1 Grain-size distributions of the 12 study sites

The $D_{50}$ calibration curve was used to produce distributed 1-m resolution grain-size maps of active channels for the 12 study

sites, from which composite GSDs of the active channel were extracted. These composite GSDs are arranged in ascending order of drainage area for inter-site comparison (Figure 11). A pattern of large-scale downstream fining clearly emerges, with an increasing proportion of fine fractions (coarse gravels) with drainage areas.  More specifically, most of the sites with drainage areas between ~30 km² and ~200 km² share very similar GSDs, with around 40 % of the active channel presenting a $D_{50}$ above 45 mm. One exception is the Béoux, showing a coarser GSD, likely related to high sediment supply from a very

active debris-flow torrent located a few hundred meters upstream of the study reach. A regular downstream fining is observed for drainage areas above 200 km², except for the Drac, showing a GSD similar to small catchment sizes. This can be related to the geological specificity of this site, compared to other study sites with large catchment sizes, which are all included in the sedimentary domain. This is not the case of the Drac, where an important part of its catchment is composed of very resistant crystalline rocks (gneiss, granites and migmatites).


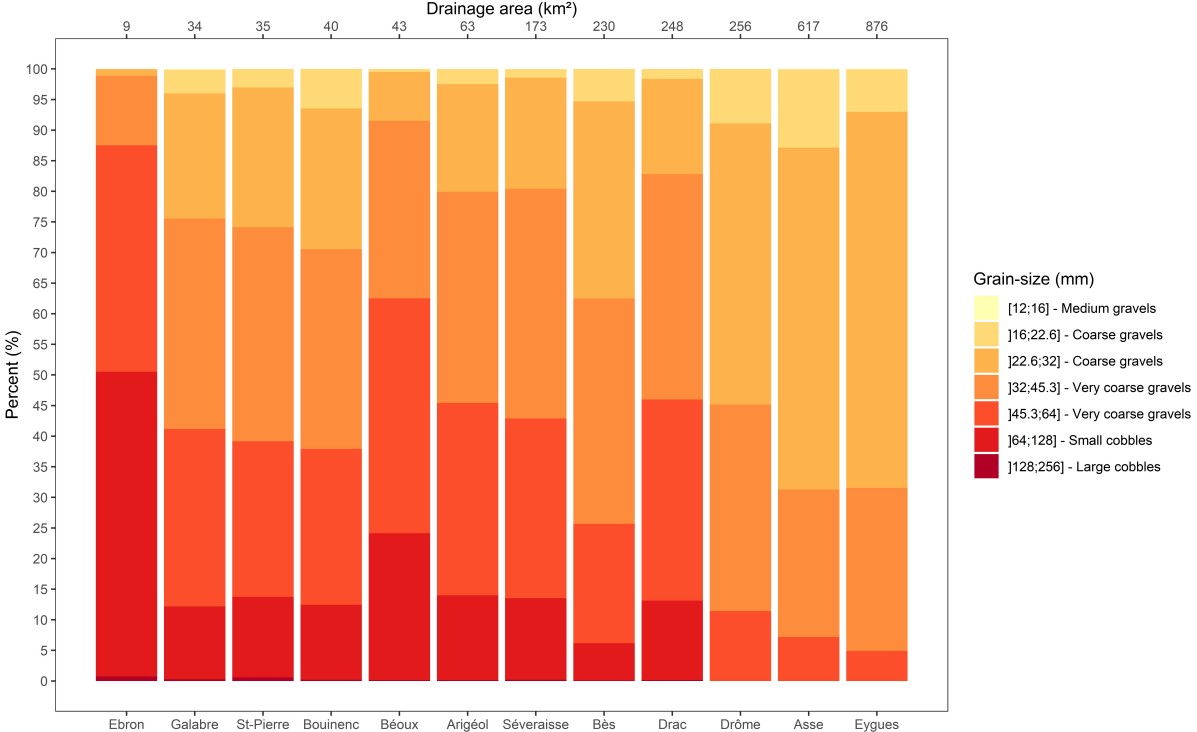

**Figure 8 : Grain-size distributions of the active channel for each study site arranged in order of increasing drainage area**



### 3.3.2 Field control of SfM-based median grain-sizes

The comparison of SfM-based (or roughness-based) and field-based reach-averaged $D_{50}$ of active channels and different geomorphic units of active channels are presented in Table 6. This comparison shows that almost all of the extracted $D_{50}$ from grain-size maps are included within the 95% confidence interval of the $D_{50}$ obtained by Wolman pebble counts. This is the case for the Arigéol, where all the SfM-based $D_{50}$ can be considered as very good estimates of "true" values observed in the field, with absolute differences always being inferior to 3 mm (less than 5% of field-based reach-averaged $D_{50}$), including

submerged portions of the active channel. A good agreement was also obtained for the investigated point bar of the Bouinenc, with an absolute difference of only 3.7 mm between field-based and SfM-based $D_{50}$ (8% of error). More contrasted results have been obtained for the Drac. Most of the $D_{50}$ extracted from the 2021 imagery are not included in the 95% confidence interval of field-based $D_{50}$. This discrepancy can be explained by an important reworking of the active channel during a series of floods that occurred in October 2023. The comparison with the 2024 imagery provides much better results, except for low-

flow channels, where an underestimation of the field-based $D_{50}$ is still observed (18% of error). However, results obtained for bars and for the whole active channels confirm a very good agreement with field-based values, with absolute differences being less than 3 mm (less than 6% of error). The pre-selection of bar pixels for exclusion of bar surfaces impacted by woody debris and fine sediment deposits does not have a strong effect on SfM-based reach-averaged $D_{50}$, whether on the Drac or the Arigéol. Differences obtained between all pixels and pre-selected pixels for these two sites are extremely small (less than 1 mm), and

do not justify any exclusion procedure for the computation of reach-averaged median grain-size of bars.

| Study sites | Morphological units | SfM-based $D_{50}$ (mm) | Field-based $D_{50}$ (mm)* |
|---|---|---|---|
| Arigéol | Bars (all pixels) | **45.3** <br> *m = 7 227* | 47.4 (43.2-51.4) <br> *n = 952* |
| | Bars (pre-selected pixels) | **45.6** <br> *m = 33 433* | |
| | Low-flow channels | **47.0** <br> *m = 23 346* | 49.7 (43.3-55.9) <br> *n = 296* |
| | Active channel | **46.9** <br> *m = 70 201* | 48.0 (44.7-51.3) <br> *n = 1 248* |
| Bouinenc | Point bar | **43.6** <br> *m = 2 256* | 47.3 (33.9-60.6) <br> *n = 120* |
| Drac | Bars 2021 (all pixels) | 46.3 <br> *m = 13 301* | 39.3 (36.0-42.6) <br> *n = 838* |
| | Bars 2024 (all pixels) | **41.4** <br> *m = 24 053* | |
| | Bars 2021 (pre-selected pixels) | 44.9 <br> *m = 168 597* | |
| | Bars 2024 (pre-selected pixels) | **41.5** | |



| | | |
|---|---|---|
| | *m = 184 231* | |
| Low-flow channels 2021 | 46.8 *m = 71 163* | 59.2 (51.7-67.6) *n = 288* |
| Low-flow channels 2024 | 48.8 *m = 67 702* | |
| Active channel 2021 | **45.7** *m = 262 266* | 43.8 (40.4-47.5) *n = 1 126* |
| Active channel 2024 | **43.7** *m = 273 689* | |

**Table 6: Comparison of $D_{50}$ obtained by Wolman samplings (field-based) and by grain-size mapping (based on the roughness proxy computed with SfM 3D point clouds) for different geomorphic units and for the whole active channels ; $m$ is the number of pixels and $n$ is the number of particles sampled; * values in brackets correspond to the 95% confidence interval of the $D_{50}$ computed with GSDtools (Eaton et al., 2019); SfM-based $D_{50}$ in bold are those included in the 95% confidence interval of the field-based $D_{50}$; the sum of pixels in bars and low-flow channels is not equal to the number of pixels in active channel, because only pixels that are fully in each morphological unit were used.**


### 3.3.3 Bars-scale morpho-sedimentary signatures

SfM-based grain-size maps have been used to explore morpho-sedimentary signatures of bars. The morphological component of the signature has been constrained by Relative Elevation Models (REMs), which were computed by subtracting at a 1-meter interval the elevation of the active channel from the averaged elevation of the main low-flow channel (i.e. the thalweg that

structured the active channel). A first example is presented at the scale of a single bar, which corresponds to the point bar of the Bouinenc that have been sampled in the field for grain-size measurement (Figure 8). A clear sedimentological pattern typical from alternate bars is detectable here. A longitudinal grain-size gradient with a downbar fining trend is clearly visible from the grain-size map (Figure 8B). Coarser grain-size patches ($D_{50} > 45.3$ mm) are preferentially located at the highest part of the bar which corresponds to its head (Figure 8B, and C). This example illustrates how classic sedimentological patterns of

gravel-bed rivers can be well-recognized on image-based grain-size maps.

A second example is provided by the exploration of the link between surface grain-size of bars and their relative elevation. This has been explored for the 3 sites where a detailed manual mapping of bars have been done (Arigéol, Bouinenc, and Drac). The distribution of the maximum $D_{50}$ of bars as a function of their 90[th] percentile of relative elevation is presented in Figure 9. Maximum $D_{50}$ were extracted from pre-selected pixels of bars (those which can be considered as non-affected by

woody debris and/or fine sediment deposits). Each of the sites shows a positive trend indicating that the relative elevation of bars has a strong effect on surface grain-size. As the coarser part of bars generally corresponds to bar heads, this result demonstrates that in many braided active channels, the coarser bar heads generally correspond to the highest bars within the active channel.






**Figure 9: Morpho-sedimentary signature of a typical point bar in the Bouinenc. (A) High-resolution SfM ortho-image (1.35 cm/pxl); (B) grain-size map extracted from imagery; (C) REM (2.7 cm/pxl); flow is from right to left; the black line corresponds to the point bar delimitation.**




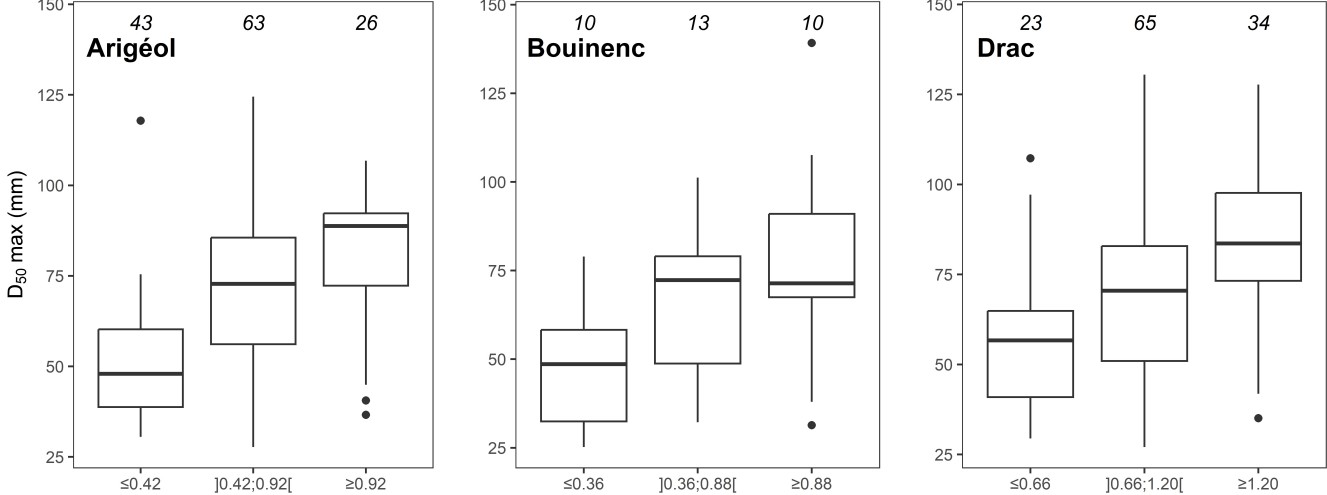

**Figure 10: Maximum $D_{50}$ of bars as a function of relative elevation; numbers of bars for which the $D_{50}$ value has been extracted are indicated on tops of the box plots; quantile distributions of the relative elevation of bars for each site have been used for discretization (≤Q25, ]Q25;Q75[, ≤Q75).**

## 470  4. Discussion

### 4.1 Photosieving with DGS

New data have been produced to test the quality of automatic particle size extractions resulting from the processing of close-up images of gravel bars. These data are crucial for the calibration of SfM sedimentological proxies. The comparative analysis of the DGS percentiles with those obtained from the manual extraction of the apparent diameters on ImageJ shows that a field

calibration of the DGS results is absolutely necessary, even for high-definition images of homogeneous sedimentary facies. A systematic bias is observed for all percentiles, with a much more pronounced shift for small percentiles. The overall performance of DGS is not as good as presented in the literature. Buscombe (2013) reported a normalized *RMSE* of 16% for $D_{50}$ using a set of 262 unconsolidated sand/gravel images. However, DGS tests on exposed Rhine gravel bars showed a much higher normalized *RMSE* for $D_{50}$, with a value of 53% (Chardon et al., 2020). This error drops to 10% after correction by linear

regression. In our case, the calibration curve of the $D_{50}$ offers a lower precision around 30%. This lower precision could be related to our calibration dataset, which is about 4 times larger than the one used by Chardon et al., (2020) on the Rhine (n = 10), and which includes 5 sites with high lithological variability. It should also be noted that our $D_{50}$ calibration curve is strongly impacted by two outliers that were collected on the Ebron, with a manually extracted $D_{50}$ greater than 120 mm. All the other points are close to the line of equality with the DGS prediction, showing that the image processing algorithm provides

a very good performance for a $D_{50}$ lower than 100 mm (nRMSE = 24%). The loss of precision above this size is probably related to the reduced number of grains in the images, which is known to have an effect on percentile predictions (Buscombe, 2013).





Our results are generally in agreement with previous tests of the DGS code showing systematic underestimation of small percentiles, and a better performance for large percentiles. This is explained by the high sensitivity of the algorithm to a small number of fine grains visible on the images (Buscombe, 2013). However, an unusual systematic positive bias for high

percentiles was observed in our calibration data. This bias could be related to a positive correlation between grain-size and shading intensity (larger grains are systematically brighter), which may lead to an overestimation of the grain-sizes predicted from the power spectral (Buscombe, 2013). This is probably the case for several images taken on sites characterized by a mixture of limestone and marly-limestone rocks (Eygues, Galabre, Arigéol, and Ebron). Limestone rocks produce larger and shinier (white) grains compared to marly limestones, which are darker and more susceptible to abrasion. However, it is difficult

to explain why this petrographic bias would only affect the large percentiles. Further studies are needed to better understand the systematic biases associated with DGS code predictions.

## 4.2 Grain-size from 3D SfM point clouds

New multi-site grain-size calibration curves for braided rivers, based on several roughness height proxies, has been produced. The best calibration curve is the one obtained for the $D_{50}$, which can be predicted with an independent error of 5 mm (14.4%

of the mean $D_{50}$ of calibration plots), using the mean roughness height derived from 3D SfM point clouds. The good performance of this roughness proxy can be explained by several sedimentological factors which are known to have a strong effect on topography-based approaches of grain-size measurement. Indeed, grain-size, shape, and imbrication are the most significant parameters controlling the grain-scale topographic variability (Pearson et al., 2017; Vázquez-Tarrío et al., 2017; Woodget et al., 2018; Wong et al., 2024). Our calibration dataset is composed of 129 sampling plots of flat and homogeneous

alluvial deposits composed of relatively coarse particles ($D_{50}$ comprised between 12.9 and 111.7 mm), with a generally low degree of imbrication, and with shapes dominated by spherical grains without many flat or elongated grains. All of these sedimentological characters, which are typical of braided alluvial deposits composed of relatively resistant rock types (e.g., limestones, gneiss, granites) are generally considered as good conditions for roughness-based approaches of grain-size (Hodge et al., 2009; Brasington et al., 2012; Rychkov et al., 2012; Vázquez-Tarrío et al., 2017). However, caution should be made

about the applicability of topography-based approaches in other sedimentological contexts, as many recent research findings have shown that image-based approach provide better grain-size proxies in river channels characterized by finer grain-sizes with a high degree of imbrication (Woodget et al., 2018; Wong et al., 2024). Another limitation of our approach is the very poor performance of the tested roughness proxies for the prediction of high percentiles (e.g. $D_{84}$). A strong data scattering has been obtained with the calibration plots, and this is likely due to a high variability of the proportion of imbricated grains among

the coarsest particles visible on sampling plots. This may explain why some sampling plots show a strong effect of the coarsest grains on the mean roughness, and some others not. Interestingly, results presented by Vázquez-Tarrío et al. (2017) showed a good performance of the roughness height not only for $D_{50}$, but also for $D_{84}$. However, the dataset was collected on a single channel reach, where we can expect a lower variability of sedimentological conditions.



Our results give insights about the transferability of the roughness-based calibration curve as the jackknife resampling
based on site exclusion shows that the vast majority of the excluded sites does not display any systematic deviation from the
calibration curve. Points remain well distributed on either side of the curve, with some exceptions. Roughness height
underestimates grain-size for Bouinenc and St-Pierre whereas it overestimates grain-size for Béoux, Drac and Drôme. Visual
check of sampling plots reveals that this is probably related to their sedimentological properties (i.e. bed-surface structures).
Grain-size underestimation for sampling plots of Bouinenc is probably due to the embeddedness of one to two cobbles in plots
dominated by pebbles. Close-range images of the Bouinenc are also affected by shading (light-related) which could lead to an
overestimation of their grain-size by DGS. Underestimation for St-Pierre may be linked with particle imbrication. Grain-size
overestimation for most of the sampling plots of Béoux, Drac, and Drôme, is probably linked to poor sorting. Indeed, when
the grain-size is not homogeneous, and the range especially large with both cobbles and pebbles, $Rh$ may be overestimated.
On the other hand, high errors in homogeneous sampling plots may be attributed to bedding. For Drac and Drôme, some
particles (coarse to very coarse pebbles) are laying on tabular surface (i.e. strata) constituted of very well imbricated and
smaller particles (medium pebbles). The difference of elevation between those particles and this flat surface may cause the $Rh$
to be higher than the 2D measured grain-size. Normalized residual errors from this jackknife resampling show that the
transferability of the calibration curve to sites that share similar physical features with those used in this study (i.e. gravel-bed
braided rivers) should enable the estimation of the $D_{50}$ with an uncertainty between 5 and 17.5 %. These values must be seen
as conservative since the global curve incorporates a greater variability than the curves tested for transferability.

### 4.3 Surface grain-size mapping

### 4.3.1 Do SfM-based grain-size maps provide good estimates of reach-averaged $D_{50}$?

Comparisons of SfM-based $D_{50}$ with field-based values obtained from intensive Wolman pebble counts revealed that our
roughness-based grain-size maps can be used for extracting a reach-averaged median grain-size of high-quality (less than 5%
of error) along several kilometers of river channels, not only at the scale of the whole active channel, but also at the scale of
different geomorphic units (unvegetated bars and low-flow channels). In the case of Arigéol, whatever the morphological unit
considered, the SfM-based $D_{50}$ is systematically within the 95% confidence interval of the field-based $D_{50}$. Interestingly, the
median grain-size computed in the submerged portion of the active channel is quite consistent with the value obtained by field
sampling. This means that uncorrected SfM 3D point clouds obtained under water provide a very good estimate of the
submerged surface grain-size. Although more contrasted results have been obtained for the Drac, this study site however
confirms a very good agreement between SfM-based and field-based $D_{50}$ for the whole active channel and unvegetated gravel
bars. If low-flow channels are considered, a substantial underestimation of the median grain-size has been observed. This
means that the submerged roughness is not always a good proxy of the grain-size, notably when water depth increases.
Although the Drac is equipped with a gauging station, it is not the case of the Arigéol, so it is not possible to compare water
discharges observed during drone flights of these two sites, that show contrasting results in terms of submerged grain-sizes.



However, water depths for both Arigéol and Drac were estimated using the SfM-DEM. Water-depth extraction was done by subtracting the minimum elevation of the channel obtained at a 10-meter interval from the elevation of the external limit of the channel (water/bar limit). The averaged water depth was 9 cm greater in the Drac (mean value of 0.35 m) than in the Arigéol (mean value of 0.26 m) at the time of drone flights. The refraction effect on SfM topography must then be greater on the Drac, thereby introducing more error into the positioning of the photogrammetric points which probably biases the prediction of the submerged grain-size.

### 4.3.2. Exploration of the grain-size effect of floods

Two sets of images have been captured for the Drac (September 2021 and March 2024), because an active hydrological period in October 2023 has complicated the comparison of the 2024 field sampling with 2021 UAV imagery. In autumn 2023, over a period of 19 days, a succession of six important rainfall events (cumulative rainfall of 456 mm) affected the catchment causing six consecutive floods. The return period estimated for the first peak flow discharge (300 m$^3$ s$^{-1}$ at a gauging station located 17 km downstream) is 10 yrs while the five others are annuals (CLEDA, 2023). Thus, a high morphological activity was maintained for two weeks in the study reach. The comparison of 2021 and 2024 grain-size maps provide some information about the sedimentological effect of this hydrological period. Contrasted trends have been observed between geomorphic units, with a 10% decrease of the bar $D_{50}$, and a 4% coarsening of the low-flow channel $D_{50}$, which cumulatively generate an overall slight decrease (4%) of the whole active channel median grain-size. Although these changes are quite moderate, they contribute to improve the comparison with field samplings, except for the low-flow channel where a significant underestimation of the reach-averaged median grain-size persists. The most important change is the decreasing of the bar grain-size, that likely results from the formation of new bars mainly composed of gravels in the vicinity of the low-flow channels.

### 4.3.3. Exploration of bar-scale morpho-sedimentary signatures

The concept of patchiness in gravel-bed rivers, known as the spatial organization of surface grain-size into patches, has been widely used in fluvial geomorphology and sedimentology to investigate textural variability of river channels (Bluck, 1976, 1979; Lisle and Madej, 1993; Guerit et al., 2014; Storz-Peretz et al., 2016). Several examples show that our SfM-based grain-size maps can provide high-resolution spatially-distributed datasets to explore patchiness of gravel-bed rivers. At a single bar scale, the example of the investigated point bar of the Bouinenc shows that a realistic grain-size sorting pattern has been provided by the grain-size map, with a classic down-bar fining, typical of alternate bars found in wandering river channels. Within the wandering Fraser River, Rice and Church (2010) found that the strongest grain-size gradient is at the bar scale. Indeed, Wolman samples showed that down-bar fining was common among compound bars with an average grain-size decrease from head to tail for 6 out of the 8 studied bars. However, the image-based grain-size mapping (robotic photo-sieving from UAV imagery) of 33 bars of a gravel-bed river in Oregon nuanced the transferability of this sedimentological pattern, with 14 out of 33 bars exhibiting a down-bar coarsening (Levenson and Fonstad, 2022). This recent study illustrates the




complexity of sedimentological signatures of bars at the scale of a single river reach, and demonstrates the great interest of high-resolution remote sensing tools for a better consideration of this complexity. Our calibrated roughness-based approach

of grain-size mapping also opens new avenues in this direction, for a more systematic observation of bar-scale grain-size gradients that can be confronted to existing conceptual or physical models of grain-size sorting.

Another example of sedimentological application is the systematic extraction of grain-size metrics of bars along a channel reach, that can be compared with morphological gradients (e.g. relative elevation). This kind of analysis along three of our study sites has shown a strong topographic effect on the maximum $D_{50}$ of bars, with a coarsening of bars with relative

elevation. Topographic level can be considered as a proxy for the magnitude of the flood that built the macroform, with lower and upper bars preferentially associated with low-magnitude and high-magnitude flow events, respectively. This is coherent with the bar-scale sorting model from Bluck (1976, 1979) showing a stage-dependent topographic sorting as supra-platform of bars (i.e. upper areas of bars, which generally correspond to the coarsest patches) are formed at high stages of floods. It has been also recognized that distinct topographic levels are often considered to create distinct sedimentological facies and

especially an upward fining succession (Miall, 2006). Consequently, maximum grain-size has been considered (instead of a more integrative mean value), in order to avoid sedimentological bias as upper bars are often wrap of decanted fine sediment deposits during the falling stage of floods. The maximum $D_{50}$ of bars is then considered here as more representative of the flow that built the bar. This hydrological imprint of the active channel grain-size patchiness is confirmed for the three investigated sites, and this demonstrates that some recent conceptual models of braided channel patchiness (e.g. Guerit et al. 2014; Storz-

Peretz et al. 2016) insufficiently incorporate hydrological forcings, and particularly the range of flood discharges that reworked the active channel.

## 5. Conclusion

In this paper, high-resolution imagery of 12 braided gravel-bed river of SE France, obtained from UAV equipped for RTK direct georeferencing, was used to develop a roughness-based grain-size calibration curve from SfM 3D point clouds.

This calibration curve was used to determine the surface $D_{50}$ with an independent error of prediction of 5 mm (14.4 % of error). A resampling procedure confirms its good transferability to braided rivers with sedimentological conditions similar to our study sites (residual prediction error of the $D_{50}$ ranging from 5 to 17.5 %). The application of the calibration curve to rasters of roughness derived from SfM point clouds of high density was used to produce distributed grain-size maps of active channels, including exposed and submerged areas, along river reaches of several kilometers. Reach-averaged $D_{50}$ values of active

channels and of different geomorphic units of the active channel (unvegetated bars and low-flow channels) obtained from these maps were in very good agreement with values measured in the field using intensive Wolman pebble counts (differences of less than 5%). This field control shows that our remote sensing approach provides a rapid, efficient, and accurate approach for determining the reach-averaged median grain-size of river reaches spanning several km in length. The potential of this methodology was also explored to characterize the grain-size patchiness of braided channels. The combination of grain-size

and topographic data shows a systematic altimetric gradient of the coarse grain-size fraction of bars, that is interpreted as an hydrological imprint. The systematic analysis of grain-size patchiness using this kind of remote sensing approach should provide new insights for the understanding of grain-size sorting processes in gravel-bed rivers, and their subsequent morpho-sedimentary signatures.

**Author contribution**

**Loïs Ribet:** Conceptualization, Data curation, Formal analysis, Investigation, Methodology, Resources, Software, Validation, Visualization, Writing – original draft preparation, Writing – review & editing. **Frédéric Liébault:** Conceptualization, Data curation, Formal analysis, Funding acquisition, Investigation, Methodology, Project administration, Resources, Supervision, Validation, Visualization, Writing – original draft preparation, Writing – review & editing. **Laurent Borgniet:** Data curation, Investigation, Methodology, Resources, Validation. **Michaël Deschâtres:** Data curation, Investigation, Methodology, 625  Resources. **Gabriel Melun:** Conceptualization, Validation, Writing – review & editing.

**Competing interests**

The contact author declare that none of the authors has any competing interest.

**Data availability**

Data will be made available on request.

**Acknowledgements**

This study has been supported by OFB (Office Français de la Biodiversité) of the French Ministry of Environment, and by Région Auvergne Rhône-Alpes (Booster R&D CAIRN). This work was performed in the framework of the LTER CNRS-INEE-ZABR (Zone Atelier Bassin du Rhône). Coline Ariagno, Benjamin Dedieu, Arthur Lopez, Franziska Losert, Sylvain Puech, and Théo Welfringer are acknowledged for the support in the field. Maxime Jaunatre and Provence Mahjoub are 635  acknowledged for their help with coding.

**Financial support**

This research has been funded by OFB (Office Français de la Biodiversité) of the French Ministry of Environment, and by Région Auvergne Rhône-Alpes (Booster R&D CAIRN).





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
