# Peer review of "Surface grain-size mapping of braided channels from SfM photogrammetry"

_EGUsphere, 2024_

## Author Comment (AC1)

Dear reviewers,
We would like to thank you for having accepted to evaluate our manuscript, and for your constructive comments. Our reply is provided below, in blue.

**REVIEWER #1**

**1. GENRAL COMMENTS**

This manuscript describes an approach for estimating median sediment sizes over a large spatial scale in multiple sections of French braided rivers. This approached is based on photographic data acquired from UAV platform.

The feasibility of such an approach has already been demonstrated in the literature. However, this new study presents some important strengths for the community, such as the large number of rivers studied (12), the study of the transferability of this method and its calibration, the evaluation of performance on different geomorphic units including water channels, and some nice examples of application.

However, there are points to be improved/discussed or possibly implemented.

- The introduction fails to clearly highlight the scientific gaps that need to be filled.
- There seems to be some information missing from the introduction that would help to really understand your objectives
- A grain size description of the reference data (manual or pseudo ground truthdata or Wolman data) is missing, it should be provided either in the methodology section or in the results section.
- information about distribution of samples surface area/$d_{max}$ area or grain number per samples could be provided instead of active chanel width in figure 1B
- a few small details may be missing in the methodology and results section. (see below the listed details by lines)
- the errors concerning the estimation of the high percentiles by the photosieving tool (here DGS) and the significant scatter for the prediction of the high percentile with the roughness calibration could be related to a problem with the surface area sampled by photosieving not being large enough (1m² for certain samples being too small), coupled with a surface area for the extraction of the roughness values which is different from that for the extraction of the pseudo ground-truth $D_{50}$. Each member of the data pair does not cover the same extent
- options to be implemented to give more weight to this study:
  - a second photosieving tool of the same type but more recent such as Sedinet (Buscombe 2020) or an oriented object detection tool could have been tested in parallel to DGS to evaluate the importance of the tool used on the subsequent performance of the $D_{50}$ estimation with roughness on this large dataset composed of 12 rivers.
  - In the same vein, a second mesh size to extract roughness could have been tested to assess the weight of this parameter.
- The error of the $D_{50}$ estimate through the roughness method seems to be accurate compared to the Wolman field data. But it is not certain that these differences in $D_{50}$ in Table 6 characterise well the performance of your method given that such a large spatial integration may include compensation.

Our reply to these general comments, which have been detailed by the reviewer in his specific comments, are provided below.

**2. SPECIFIC COMMENTS**

**Introduction**
The introduction presents the history of the progress of the techniques available to characterize the grain size distribution of braided rivers. Nevertheless, it seems that some recent publications on the subject are not presented here but appear later in the paper.
The introduction fails to clearly highlight the scientific gaps that need to be filled. For sure your work is revelent. You just need to reshape the way you present your objectives..

I don't understand exactly what the main objective of this paper is:
- To evaluate the performance of large-scale spatial estimation of $D_{50}$ using an RTK-equipped drone (but GCPs are ultimately used, and the errors are not compared with point clouds constructed without the use of RTK during photo acquisition).
- Evaluation for a sedimentological application? (not clear).
- It is the first examples of application of such a method in this field? (in relation to the results obtained in section 3.3.1 and 3.3.3 ).

I think your first 2 objectives should be reworded. Perhaps there aren't 4 but 3?
- The first one mentioned doesn't seem to be an objective in itself. You are evaluating the specific error of the DSG tool for photosieving. Studies already exist on this subject. In reality, I think you want to correct the 'systematic' errors of your automatic photosiving to then ensure that you minimise the errors of your $D_{50}$~roughness calibrations. You do this because you know from your literature reviews that automatic photosiving methods are biased. But the problem is that you never talk specifically about DGS in your introduction, so it's complicated for you to introduce, and for the reader to understand, your 1st objective. Eventually you could have had this objective if you had compared several photosieving tools (at least two). In the end, I would say that either this is a sub-objective of your objective 2, or it could be an objective 1 called 'to evaluate the automatics phototosieving error and developpement of correction with field-base calibration'.
- Perhaps objective 2 should be reformulated. It seems to be strongly linked to the use of RTK, but then neither the results nor the discussions refer to RTK.

Although this comment on objectives is not shared by the second reviewer, who considers that "the research questions are excellent", a revision of the last paragraph of the introduction has been done, to better present the addressed scientific questions. We agree that the first objective (DGS performance testing) is more a methodological step than a research objective, and it has been deleted. The second objective (reach-scale application of SfM grain-size mapping) has been reformulated to better present the addressed challenge of grain-size mapping at the reach scale. The last two objectives have been conserved.

In **L 35 to 49 or L 87 to 10**: you should clearly introduce the fact that it may be necessary to correct the data obtained by photo-sieving with a calibration.
This is now specified in the second paragraph of the introduction.

**L39**: The list of photosieving solutions could be extended to include more recent solutions (ex: Buscomb 2020), the latest of which is already 11 years old.

This pertinent reference has been added.

**L42**: "with typical measurement errors of ...". Could you specify the error of what, b-axis? number of grains? what type of error, bias? RMSE?
This is now specified (random measurement errors of grain-size percentiles).

**L90**: rather strange to introduce Basegrain when you don't use it at all in your workflow. In addition, Basegrain is more of an object detection type, whereas DSG is not. Perhaps part of the methodology section 2.2.2 should be reworked in the introduction.
BASEGRAIN is mentioned here because it is the code that has been used in the robotic photosieving approach developed by Carbonneau et al. (2018), that we found useful to cite in the state of the art of remotely sensed grain-size approaches.

**L106/107**: "but, it has never been tested…". I recommend that authors consult the publication by Mair et al (2022) (https://doi.org/10.5194/esurf-10-953-2022) which evaluated the variations in performance when using UAVs equipped with RTK or GCP for sedimentological purposes.
Thank you for for this recommendation. The text has been changed, and the reference is now cited here.

**Methodology**

The methodology section is very well structured. However, the description of the active width of the channel, although obviously very interesting, does not seem to have a place in this paper since it is never included in the results or the discussion.
It might be interesting to use this space to describe the particle size distribution of the study sites. It seems unthinkable to carry out this study concerning the estimation of the $D_{50}$ without giving any information on the particle size in the presentation of the study site, not even in one column of a table.
It is right that the mean active channel width of the study reaches is not used in results and discussion. However, we think that it is important to keep this information in the presentation of the study sites (and in Tab. 1), not only because this basic descriptor is an essential scaling metrics of braided channels, as important as the drainage area, but also because it has been used to calculate the normalized active width (W*), which is considered as a good proxy of braided channel sediment regime, as explained in the second paragraph of section 2.1. This allows us to show that our grain-size mapping approach has been calibrated with a dataset covering braided channels with contrasted sediment supply conditions. This is now better specified in the discussion. We think that information on channel width is important for the contextualization of our research findings.
Providing a representative information of grain-size distributions of the 12 study reaches is not an easy thing, because this implies expensive and time-consuming field surveys, that can be estimated at 48 person-days. This is notably why the development of remote sensing approaches for surface grain-size extraction is needed, and our study is part of this objective. Intensive Wolman pebble counts have been undertaken for 2 sites (Arigéol and Drac), as presented in the methodological protocol (see section 2.3.4), but this was done for comparison purposes with SfM-based grain-size estimates. Surface grain-size distributions derived from SfM products of the 12 study sites are provided in the results (Fig. 12), but it would be quite inappropriate to use this information in the presentation of the study sites, because this is a research result, and not a preliminary information. The best that we can provide in the presentation of the study sites is a qualitative identification of the dominant grain-size, based

on a visual inspection of ortho-images that have been produced for this study. Table 1 has been completed with a new column about the dominant grain-size, with information in the caption about this new preliminary descriptor. Pictures provided in Fig. 2 C and D also allow readers to figure out grain-size ranges of the study sites.

**L75/76**: "Obtained images..". How were these images scaled? Do they come from SfM processing (orthophotos)? Or do they come from simple photographs taken with the drone, then scaled using the meter placed next to the frame? is this step performed in a GIS environment? How many georeferencing points were used? 2? 4 points?
These close-range images were only used to obtain pseudo-ground truth data with DGS, and they have not been processed with Metashape. They have been scaled with the rule attached to the wooden frame, as explained in the caption of Fig. 2, as well as in the previous paragraph.

**L180**: "Digital Grain-size (DGS) code developed by Buscombe (2013)," It is curious that this DGS tool is not referenced in the introduction, whereas BASEGRAIN is while not been utilized in this study. Additionally, Sedinet, developed by the same author as DGS (Buscomb, 2020), but a more recent tool, is not referenced in the introduction either. I guess the reason is that Chardon et al (2020) have shown that DGS performs better than Sedinet. This could be added somehow in the introduction.
DGS is implicitly mentioned in the introduction, since Buscombe 2013 is cited in the paragraph dealing with photosieving. However, it is now mentioned explicitly in the introduction. Results from Chardon et al. (2022) showing a better performance of DGS as compared to SediNet, which is the reason why DGS has been chosen here to produce pseudo ground-truth data, are now explicitly mentioned in the methodology (section 2.2.2). NB: the paper by Chardon et al. (2020) does not provide any comparison between DGS and Sedinet, and we assume that the reviewer was referring to results presented in Chardon et al. (2022).

**L189**: "images were corrected...". Can you indicate how, with which program you are carrying out this phase, which seems to be important for the rest of your study?
Images were corrected with ImageJ. This is now specified in section 2.2.2.

**L199**: "At least one hundred .." What is the distance between the nodes? The number of nodes will be very small to avoid sampling the same grain several times. Could you give us some details? It might be useful to quickly mention the range of the number of grains sampled and inter distance node.
Distance between the nodes is around 10 cm for 1 $m^2$ plots, and 20 cm for 4 $m^2$ plots. Grid spacing is chosen to maximize the number of grain while avoiding sampling the same grain twice. This is now specified in the text. The number of grains is comprised between 100 and 110. This protocol of on-screen manual extraction of b-axis has been proposed by Graham et al. (2005a), and it is commonly used to generate pseudo ground-truth datasets (Barnard et al. 2007; Buscombe 2013; Chardon et al. 2022).

**L203/204**: "several studies demonstrated that this systematic error is low for grid-by-number sampling".As if the systematic error in grid-by-number sampling were small compared to something else. But what is that other thing?
According to the cited papers addressing the fabric error from photosieving (Graham et al. 2005b, Dugdale et al. 2010), b-axis measured on images were directly compared with field measurements of surface grains using the paint-and-pick approach. Readers can easily find this information by consulting the cited papers. This is now specified in the text.

**L259**: "The radius…". Why extract the roughness in a sphere rather than in a 'box' ? A box covering exactly the same area as the photosieving data could be more suitable for accurately assessing the deviation between the SfM data and the photosieving data? Why use a radius of 0.5m on the SfM data if the calibrations are obtained from photosieving data covering at least 1m² ?

There is a confusion here about the method that has been used for the roughness measurement. The roughness extraction tool available in CloudCompare is based on a sphere of predetermined radius that is used for each point of the 3D point cloud to compute a best fitting plane around that point, and the roughness height is computed as the shortest vertical distance between the point and the best fitting plane. As the roughness height is calculated for all the SfM points contained in the calibration plots, roughness measurements and grain-size percentiles derived from DGS are extracted for exactly the same surface of calibration. All the details concerning the roughness height measurement with CloudCompare are available in the user guide of the software. This is a commonly-used protocol of roughness extraction, that has been explained in several recent papers that are cited in the manuscript (Vazquez-Tarrio et al. 2017, Woodget and Austrums, 2017, Chardon et al. 2020).

In the first version of the paper, the radius for roughness computation was fixed at 0.5 m. This value was chosen following results presented by Vazquez-Tarrio et al. (2017) on a braided reach with a grain-size distribution quite similar to our study reaches. An optimum value of 0.5 m was obtained in this study. However, following this reviewer's comment, a sensitivity analysis was undertaken to determine the optimal value of the radius. An initial value of 0.1 m was chosen, and an iterative procedure was done by incrementing the radius size by 0.1 m up to a maximum value of 1 m. The optimum value was determined as the one presenting the best correlation ($R^2$) between D50 and the mean roughness height. The optimal value range between 0.4 and 0.5 m, and the radius was fixed at 0.5 m. This is now specified in the text, and a new figure presenting the sensitivity analysis was provided in the supplementary material (Fig. S1).

**L262/263**: "all theses predictors..". It is not clear whether the $D_{50}$ used in these regression analyses is the raw photosieving $D_{50}$ or the $D_{50}$ corrected by the previous calibrations from section 2.2.3 . This should be clearly stated.

The corrected D50 has been used. This is now clearly specified.

**L301**: "No major morphological changes occurred..". Totally understandable that no campaign was carried out in 2024. Having said that, it would have been interesting to have a new data acquisition for the Arigéol site to assess the differences between the two sets of data (2020/2024) for a similar expected result (the precision, the reproductibility of the method), in comparison to Drac where $D_{50}$ estimate are expected to be significatively different.

We agree that a 2024 UAV survey in the Arigéol would have been useful for giving more strength to the reproducibility of our results, but we had to save time and resources for addressing other topics included in the supporting scientific project.

**Results**
The section is well structured.

**L324/325**: "Median of the computed...". error compaire to manual data may simply due to sampling area being too small (1 m² for these patches?) to assess the entire distribution correctly, as there are too few coarse grains to characterize them correctly. In addition, using only the grains present on grid nodes and not all the grains present in the photo amplifies this

phenomenon. The error may not be so important, but may apear large due to the method used to acquire and process the reference data.

As mentioned in the following paragraph, these two outliers correspond to large sampling areas (4 m²), where the number of grains is above the threshold below which DGS performance can be considered as limited (250 grains according to Buscombe 2013). We do not think that grid sampling exacerbate the error, because this is the commonly-used approach for the production of ground-truth data to control the quality of DGS percentiles. As mentioned in the original DGS paper (Buscombe 2013), the wavelet-based approach has been specifically designed to produce a grid-by-number grain-size distribution. This is explicitly mentioned in the abstract of this paper: "The grain-size distribution obtained is equivalent to the distribution of apparent intermediate grain diameters, grid by number style".

**L336**: "Although the number of grains…". Another recommendation is the use of a sampling area equal to 100 times the area of the largest grain (Petrie and Diplas, 2000). Instead of chanel width, the distribution of the ratio sample area/$d_{max}$ area or the grain number per samples could be provided. After an explanation concerning acceptable area sampling or grain number samples, I think you could simply present the error results and establish roughness calibration without these 2 outliers, These errors don't seem to be linked to photosieving, but to the reference data obtantion..

Thinking about it, it might be appropriate to use a a roughness extraction area 100 time greater than the expected $D_{max}$ to extract the roughness.

It is assumed here that the reviewer refers to earlier works of Diplas and Fripp (1992), since the theoretical work from Petrie and Diplas (2020) showed that sample areas smaller than 100 times the Dmax can be acceptable for characterizing the median grain-size. Area-based sample size recommendation (>100 times the $D_{max}$ area) from these works, but also from more recent studies (Graham et al., 2010), have been proposed for areal sampling, and we are not sure that these recommendations should be applied to grid-by-number sampling. Petrie and Diplas (2020) have notably shown that the number of grains required for an areal sampling is much greater than for a grid-by-number sampling, to achieve a given precision of percentiles. The important criterion for the grid-by-number sampling is the grid spacing, that should be greater than the $D_{max}$. Therefore, instead of considering the ratio between the sampling area and the area of the largest particle, we have considered the ratio between the grid spacing (distance between nodes) and the b-axis of the largest grain on each DGS calibration plot (n=44). This criterion presents a mean value of 0.96, and is comprised between 0.41 and 2.70. Calibration plots with low values of the ratio, that should be considered as the most problematic data points, are in fact well distributed along the equality line (Fig. 1). It is also clear that this ratio cannot be used to justify the suppression of the two outliers, as suggested by the reviewer. Therefore, it is not possible to conclude that the grid spacing and the size of the calibration plots are both too small for the calibration of DGS percentiles in our investigated range of grain-sizes. We agree with the reviewer that larger calibration plots could have been of interest for improving the quality of grid-by-number percentiles measured with ImageJ, and particularly for the quality of coarse percentiles (D84, D90), but this would have been quite unpractical in the field, simply because homogeneous alluvial surfaces (without woody debris, vegetation, and silt or sand deposits) of large size are not easy to find. This could also have a negative effect on DGS results, with the integration of non-alluvial objects on images. We believe that the two sizes of our calibration plots (1 and 4 m²) are a good compromise for our field sites, and they are in good agreement with previously reported grid-by-number photosieving studies (1.2 m² in Graham et al., 2005a, 1 m² in Chardon et al., 2020). However, a specific analysis of the optimum sampling area (with

respect to $D_{max}$) for grid-by-number photosieving would be quite useful. The text (both results and discussion) has been completed to integrate this point.

[Figure]

Figure 1. Effect of the ratio between the grid sampling size and the b-axis of the largest grain on the relation between DGS-computed D50 and manually measured D50 (with Image J); the dotted line corresponds to the equality line

**L360/361**: "The high data scatter". This continues to make me think that perhaps these errors for the high percentiles are linked to the too small size of the surface area photographed for some samples and the size of the mesh used to extract the roughness data, which are also not the same..
See our replies to previous comments about the roughness computation (L259) and size of sampling plots (L336).

**L379**: "The residuals prediction..". We don't know where to look. Could be changed by: "The **standard error of the prediction residuals** varies from 5 to 17.5%.
Done

**L354**: "The $D_{50}$ calibration curve…". I suppose that the calibration used to map $D_{50}$ spatially is: $D_{50} = 1.9\ Rh + 12$ (according to table 4) for all the sites. Could you specify this in the text ?
This is now specified.

**L396**: Change Figure 11 by **Figure 8**
Done

**Figure 8**: very nice output from your workflow
Thank you

**L446, 448, 449**: change Figure 8 by **Figure 9**
Done

**Figure 9**: Very interesting result. I wonder how it can be done quickly on more sites. I have the feeling that a limiting factor is the precise delimitation of the area to be mapped in the end, the exclusion of vegetated areas, woods...
We appreciate this positive comment. We will see if an additional example can be provided in the supplementary material, in order to avoid increasing too much the size of the manuscript.

**L454**: Change Figure 9 by **Figure 10**.
Done

**Discussions**

The discussions section is well structured. The general comments at the beginning and the points listed below (sometimes linked) can at least be discussed.

**L476**: "shift for small percentiles". Do you have any explanation?
This negative bias of DGS for small percentiles was already shown by Buscombe (2013), who explains this by the high sensitivity of the wavelet approach to few numbers of the smallest grains on images. This is now mentioned in the discussion.

**L480**: I wonder what the errors might have been if you had removed the 2 outlier patches and also used another photosieving tool as Sedinet for examples instead of DSG. This is just a question that doesn't negate your study, of course.
This is already specified in the results (second paragraph of section 3.1). When the two outliers are removed, the error (normalized RMSE for D50) drop from 30% to 24%. Concerning SediNet, see our reply to previous comment (L180).

**L485**: "a very good performance" Perhaps 'Very' is excessive, as there is still a 24% error. How many % of error must we accept to consider a result as just 'good' (50% error)?
You are right, "very good" was replaced by "good".

**L495**: "Further studies are needed …" I still think it would have been useful to use at least one other tool to assess whether better calibration with roughness is possible. And also use another radius or "box area" for extracting roughness value.This kind of analyses could have given clues to indicate limiting factors and theire weight.
Perhaps using one or more other tools would have given you D$_{50}$ estimation results similar to your current DGS results, Then you could have conclude that there's no point in using more sophisticated tools. I see this as an option for improvement (not mandatory) which would add interest to your paper for the community. Your work seems perfectible.
As previously said in our reply, the best performing tool (DGS) from the study by Chardon et al (2022) has been chosen for our calibration dataset, and it was not in the scope of this study to test all of the available photosieving tools, notably because they are many. However, we agree with the interest of addressing this topic, especially because very recent developments in using machine learning for grain-size application have been published (e.g. the GALET code, Mörtl et al., 2022). We are now testing the quality of grain-size extraction with this tool, using our grain-size dataset, but this is a work in progress that will need time to produce communicable results.
Concerning the radius for roughness extraction, the paper is now completed with a sensitivity analysis (see our reply to comment L259).

**L506**: "and whith shapes dominated by spherical", Is this your opinion or concrete facts? How did you quantify this ( imbrication, grain shape). Did you measure a sample to confirm this?

This is a qualitative judgment based on visual inspection of the 129 calibration plots used in our study. This is also supported by the dominant rock types that are found on the study sites, which correspond to hard rocks (limestones, gneiss, sandstones…) producing spherical grains. This is indicated in Table 1 and in the presentation of the study reaches. We did not calculate any shape or imbrication metrics. However, images of the calibration plots have been now provided as supplementary material to support this (Fig. S6).

**L512 to 518**: "Another limitation…" Again, error on the highest percentile may initially be due to a difficulty in correctly characterising the end of the distribution because the calibration area is too small. To refute this hypothesis, no data is available. No presentation of the particle size distribution of the study reaches is given in the material and methods section, even though this is precisely what this study seeks to predict. It would also be interesting to know the distribution of the ratio area photographed/ $D_{max}$ area. Furthermore, you explain these errors by mentioning a high degree of imbrication variability when 8 lines above you talk about homogeneous patches with a low degree of imbrication. It's a bit contradictory, or maybe you need to rework this paragraph.

A generally low degree of imbrication is mentioned for the whole population of grains that has been captured with our calibration plots, but when only coarse grains are considered, a strong imbrication may occur for some specific large grains. This is not really contradictory. We however agree with the idea that the area of calibration plots could be a factor explaining the poor performance of the roughness approach for the highest percentiles. The text has been completed to integrate this point.

**L538/541**: You are presenting a method for estimating the $D_{50}$ with a high resolution (1m²) but the performance (or error) is assessed at an extremely low resolution (Wolman over a distance of at least 800m). How is the average $D_{50}$ obtained by Wolman  representative of the reality on sites? What is the standard deviation, for example, along the different transects? Your quantification of the errors aggregates so many factors that it seems difficult to really isolate the quality of the $D_{50}$ estimates based on roughness and may results from compensation.  Wouldn't it have been better to characterize each of the sedimentary unit in detail with 2/3 field samples per unit, and then compare per unit the $D_{50}$ estimated using the roughness extracted from these same locations? Of course, this doesn't seem feasible at this stage.

We agree that the sedimentary unit (e.g. gravel-bar) is the relevant spatial unit for assessing the local error of the high-resolution grain-size map, and this has been done, but for just one sedimentary unit (the gravel-bar of the Bouinenc). However, we believe that there is a real interest to show how good are the estimates of the mean active channel D50 that can be obtained with our high-resolution grain-size maps. This is of uppermost importance for many applications, like bedload transport modelling, 1D hydraulic modelling, or simple comparative analysis of different channel reaches. We also believe that our intensive Wolman pebble counts provide a representative reach-averaged mean D50. We have no reason to think that our sampling strategy presents some important bias, as all the different sedimentary units have been sampled in proportion to their relative surface (systematic sampling at regular interval along 10 cross-sections placed across the entire active channel, see cross-sections maps provided in the supplementary material). This is a common sampling scheme for obtaining a reach-averaged GSD (referred to as the "spatially integrated sampling scheme in Bunte and Abt, 2001). The total number of sampled grains is > 1000 for each of the two

channel reaches, and 95% confidence intervals of the D50 have been computed using a specifically designed statistical tool (GSDtools from Eaton et al. 2019). However, a complementary statistical analysis of the representativity of Wolman samples has been implemented, using the approach presented by Mosley and Tindale (1985) on a braided river in NZ, using data quite similar to ours. This analysis is presented in a new table (Table 7) where several parameters are presented (including standard deviations of D50 values obtained for each cross-section). It reveals margins of errors of reach-averaged D50 comprised between 10 and 20%. These new results are used to strengthen the discussion of the comparison between SfM-based and field-based reach-averaged grain-sizes (section 4.3.1). Although it is not possible to exclude a compensation effect, our complementary analysis allows to better characterize the uncertainty of field data based on grain-size heterogeneity along the study reaches, and the discussion has been completed by mentioning the possibility to conduct a stratified field sampling based on sedimentary facies. However, this kind of approach would have been quite impractical for our study reaches, where active channels are composed of a great variety of sedimentary facies.

**L541/547**: "Interestingly …underestimation of the median grain-size has been observed". You mention a "very good" estimate of the $D_{50}$ in submerged areas, with a "substantial" underestimate. That may be the case for the Arigéol site (around 5%) but it's absolutely not the case for DRAC, with at least a 17% difference. I don't know what you define by the terms 'substantial', 'good' and 'very good' in terms of percentage, but I think they should be avoided.
These terms have been removed from the text.

**L538/556**: This paragraph is a bit messy, it could be reworded with something like this:
Comparisons of SfM-based D50 with field-based values obtained from intensive Wolman pebble counts revealed that our roughness-based grain-size maps can be used for extracting a reach-averaged median grain-size of high-quality (less than 5% of error) along several kilometers of river channels, not only at the scale of the whole active channel, but also at the scale of different geomorphic units (unvegetated bars and low-flow channels). **Nevertheless**, the median grain-size computed in the submerged portion of the active channel present more contrasting results, **respectively -5.5 and -17.5%** for the Argéol and Dracs site. If low-flow channels are considered, underestimation of the median grain-size has been observed and may increase when water depth increase. Water-depth have been evaluated by subtracting the minimum elevation of the channel obtained at a 10-meter interval from the elevation of the external limit of the channel (water/bar limit). The averaged water depth was 9 cm greater in the Drac (mean value of 0.35 m) than in the Arigéol (mean value of 0.26 m) at the time of drone flights. The refraction effect on SfM topography must then be greater on the Drac, thereby introducing more error into the positioning of the photogrammetric points which probably biases the prediction of the submerged grain-size This means that the submerged roughness is not always a good proxy of the grain-size, notably when water depth increases.
The paragraph has been reworded as suggested.

**Citation**: https://doi.org/10.5194/egusphere-2024-3697-RC1

**REVIEWER #2**

General comments:
This is a very good manuscript, whose topic is highly appropriate for the readership of ESurf. The introduction is thorough and an excellent review of the state of science in this field/technique to mapping sediment size. The research questions are excellent (L105-111) – they are well defined and progress this field. The authors have collected and make use of an extensive dataset to investigate whether surface grain size can be mapped using point cloud data from SfM photogrammetry. It is great to see that 12 different rivers were used to calibrate the roughness metric. The development of this metric is rigorously undertaken and statistically informed. However, the paper isn't just confined to a technical or methodological contribution, it also makes use of the proxies to interpret grain size patterns on bars, which lead to the observation of trends (e.g. L448-449l; L455-6), which is nice. Overall, this manuscript contributes to wider efforts to create digital models of riverscapes and to enable analysis of spatially continuous data. I agree with the observation on L585 that the findings may enable existing models of grain-size sorting to be verified or adjusted – I'd encourage pursuit of this! I recommend publication, subject to addressing the comments below.
Thank you for this positive general comment, that is much appreciated.

Specific comments:
1. Thorough check for consistency in how you describe the different study areas. L159 they are called "study reaches" and L163 you say "sites". If different phrases are needed for different things then clearly define these.
   "site" was replaced by "study reach" throughout all the manuscript, for consistency.
2. What is the bias that is introduced to "real world" (i.e. to the bar scale) application of the techniques when calibration plots were chose to be flat and (especially) homogenous.
   This is a good point that needs to be better considered in the limitations of our approach. However, the fact that the calibration curve was obtained from flat and homogeneous surfaces does not seem to have a strong effect on D50 extraction, as shown by the comparison of field-based D50 obtained at the bar scale (Bouinenc data, see Fig. 9 and Table 6), or even at the reach scale. This point is now mentioned in the discussion.
3. Can you explicitly clarify whether you map grain size on just dry surfaces or whether you can also apply this to wet surfaces; can you discuss this in the discussion? I note that L548 mentions this but it's not clear in the methods/results.
   Grain-size maps were produced for dry and wet surfaces. This is now explicitly mentioned in the methods and results section. This point is already addressed in the discussion (section 4.3.1).
4. Is a figure to show temporal trends in grain size (2020 to 2024) missing – it is mentioned on L563?
   The comparison is based on extracted mean D50 from 2020 and 2024 grain-size maps, but not on a differential map. This is now better explained in the text to avoid confusion.

Technical comments:
L16 Abstract : Can you be more specific about what a "proxy" is for a potentially general abstract reader?
Done

L57 Insert "the" before Feshie
Done

L98 Insert "sediment on" before gravel
Done

L100 Insert "an" before approach
Done

L178 Were shadows avoided when acquiring images? Overcast conditions? If there were shadows, what are the implications in point cloud reconstruction?
To avoid shadowing effects, UAV surveys were taken in the late morning or early afternoon, as much as possible. This is now specified in the methods (section 2.3.1, UAV surveys). However, shadows do not really have a strong effect on SfM 3D point clouds, as can be seen on the Bouinenc gravel bar example provided in Fig. 9, where a very good grain-size information is produced in areas affected by shadows, as you can see on the ortho-image.

L223 You may be interested in Stott et al (2020) who, in one of their experiments, assessed the accuracy of using this DEM with the same field set up as you https://doi.org/10.3390/drones4030055
Thank you for this interesting reference. This paper is now cited in the introduction section dealing with direct georeferencing.

Table 2 It would be useful to present the range of the point cloud density for each site. Point cloud density can be uneven, so an average isn't always very useful.
The range of point density (D25-D75) is now included in Table 2.

L300 Change "samplings" to "samples"
Done

L303 What does "quite stable" mean? Can you rephrase/clarify
"quite stable" was replaced by "unchanged"

Figure 9 Is the grain size shown on the legend D50?
Yes, this is now specified in the caption

Section 4.1 Reid et al (2019) – already cited – developed a relationship between roughness and D50 that included bars with coarse grains (D50 > 200 mm), if you need further references on investigations that have also included large grains in the calibration.
This reference has been already cited in introduction as an example of roughness-based approach for grain-size extraction. However, it is not really appropriate to consider this paper in this part of the discussion, since DGS was not used for producing ground-truth data. Roughness proxy was directly compared to D50 obtained from field-based Wolman pebble counts in this study.

**Citation**: https://doi.org/10.5194/egusphere-2024-3697-RC2

**References**

Barnard, P.L., Rubin, D.M., Harney, J., Mustain, N., 2007. Field test comparison of an autocorrelation technique for determining grain size using a digital 'beachball' camera versus traditional methods. Sedimentary Geology, 201(1-2), https://doi.org/180-195. 10.1016/j.sedgeo.2007.05.016

Bunte, K. and Abt, S.R., 2001. Sampling surface and subsurface particle-size distributions in wadable gravel- and cobble-bed streams for analyses of sediment transport, hydraulics, and streambed monitoring. RMRS-GTR-74, USDA Forest Service, Rocky Mountains Research Station, General Technical Report 74.

Buscombe, D., 2013. Transferable wavelet method for grain-size distribution from images of sediment surfaces and thin sections, and other natural granular patterns. Sedimentology, 60(7), 1709-1732. https://doi.org/10.1111/sed.12049

Carbonneau, P.E., Bizzi, S., Marchetti, G., 2018. Robotic photosieving from low-cost multirotor sUAS: a proof-of-concept. Earth Surface Processes and Landforms, 43(5), 1160-1166. https://doi.org/10.1002/esp.4298

Chardon, V., Schmitt, L., Piégay, H., Lague, D., 2020. Use of terrestrial photosieving and airborne topographic LiDAR to assess bed grain size in large rivers: a study on the Rhine River. Earth Surface Processes and Landforms, 45(10), 2314-2330. https://doi.org/10.1002/esp.4882

Chardon, V., Piasny, G., Schmitt, L., 2022. Comparison of software accuracy to estimate the bed grain size distribution from digital images: A test performed along the Rhine River. River Research and Applications, 38(2), 358-367. https://doi.org/10.1002/rra.3910

Diplas, P., Fripp, J.B., 1992. Properties of Various Sediment Sampling Procedures. Journal of Hydraulic Engineering, 118(7), 955-970. https://doi.org/10.1061/(ASCE)0733-9429(1992)118:7(955)

Dugdale, S.J., Carbonneau, P.E., Campbell, D., 2010. Aerial photosieving of exposed gravel bars for the rapid calibration of airborne grain size maps. Earth Surface Processes and Landforms, 35(6), 627-639. https://doi.org/10.1002/esp.1936

Graham, D.J., Reid, I., Rice, S.P., 2005a. Automated sizing of coarse-grained sediments: Image-processing procedures. Mathematical Geology, 37(1), 1-28. https://doi.org/10.1007/s11004-005-8745-x

Graham, D.J., Rice, S.P., Reid, I., 2005b. A transferable method for the automated grain sizing of river gravels. Water Resources Research, 41(7), 1-12. https://doi.org/10.1029/2004WR003868

Graham, D.J., Rollet, A.-J., Piégay, H., Rice, S.P., 2010. Maximizing the accuracy of image-based surface sediment sampling techniques. Water Resources Research, 46. https://doi.org/10.1029/2008WR006940

Mörtl, C., Baratier, A., Berthet, J., Duvillard, P.A. and De Cesare, G., 2022. GALET: a deep learning image segmentation model for drone-based grain size analysis of gravel bars. In: M. Ortega-Sànchez (Editor), Proccedings of the 39th IAHR World Congress, 19-24 June 2022. IAHR, Granada, Spain, pp. 5326-5335.

Mosley, M.P. and Tindale, D.S., 1985. Sediment variability and bed material sampling in gravel-bed rivers. Earth Surface Processes and Landforms, 10(5): 465-482. 10.1002/esp.3290100506

Petrie, J., Diplas, P., 2000. Statistical approach to sediment sampling accuracy. Water Resources Research, 36(2), 597-605. https://doi.org/10.1029/1999WR900321

Vázquez-Tarrío, D., Borgniet, L., Liébault, F., Recking, A., 2017. Using UAS optical imagery and SfM photogrammetry to characterize the surface grain size of gravel bars in a braided river (Vénéon River, French Alps). Geomorphology, 285, 94-105. https://doi.org/10.1016/j.geomorph.2017.01.039

Woodget, A.S., Austrums, R., 2017. Subaerial gravel size measurement using topographic data derived from a UAV-SfM approach. Earth Surface Processes and Landforms, 42(9), 1434-1443. https://doi.org/10.1002/esp.4139